# The design, deployment and testing of Kriging models in GEOframe with SIK-0.9.8

Marialaura Bancheri[1], Francesco Serafin[1], Michele Bottazzi[1], Wuletawu Abera[3], Giuseppe Formetta[2], and Riccardo Rigon[1]

[1] Department of Civil, Environmental and Mechanical Engineering, University of Trento, Italy
[2] Centre for Ecology & Hydrology, Crowmarsh Gifford, Wallingford, UK
[3] International Center for Tropical Agriculture (CIAT), P.O.BOX 5689, Addis Ababa, Ethiopia

*Correspondence to:* Marialaura Bancheri (marialaura.bancheri@unitn.it)

**Abstract.** This work presents a software package for the interpolation of climatological variables, such as temperature and precipitation, using Kriging techniques. The purposes of the paper are: (1) to present a geostatistical software that is easy to use and easy to plug-in to a hydrological model; (2) to provide a practical example of an accurately designed software from the perspective of reproducible research; and (3) to demonstrate the goodness of the results of the software and so have a reliable alternative to other, more traditional tools. Eleven types of theoretical semivariograms and four types of Kriging were implemented and gathered into Object Modelling System compliant components. The package provides real time optimization for semivariogram and kriging parameters. The software was tested using a year's worth of hourly temperature readings and a rain storm event (11h) recorded in 2008 and retrieved from 97 meteorological stations in the Isarco River basin, Italy. For both the variables, good interpolation results were obtained and then compared to the results from the R package, *gstat*.

## 1 Introduction

Meteorological forcing data such as rainfall, temperature, and solar radiation are the dominant controlling factors for the hydrological cycle, energy balance and ecosystem processes (Ly et al., 2013). These data, besides being important of themselves, are the natural input to distributed and semi-distributed hydrological models. Their quality and precision affect the accuracy of results, (Xu and Singh, 1998; Stooksbury et al., 1999; Balme et al., 2006; Abera et al., 2017). In fact, all the surface water models require a reliable precipitation dataset that is complete both in space and in time. Quite often, however, datasets of hydrological variables suffer from errors and missing data, therefore, filling the gaps in time series by estimating the missing values is a common approach to solving this problem, (Eischeid et al., 2000; Saghafian and Bondarabadi, 2008; Di Piazza et al., 2011; Adhikary et al., 2015). Several algorithms for the spatial interpolation of meteorological data are available in the literature: Thiessen polygons (e.g., Thiessen, 1911; WMO, 1994), inverse distance methods (Ly et al., 2013), interpolation with splines (e.g., Mitášová and Mitáš, 1993; Hutchinson, 1995), Kriging (e.g., Krige, 1951; Matheron, 1981; Goovaerts, 1997) and other types of interpolation (e.g., Robeson, 1992; Li and Heap, 2011, and references therein). Their performances have been assessed by several authors, among others Tabios and Salas (1985) and Jarvis and Stuart (2001), who have concluded that Kriging is one of the best techniques for the spatial interpolation of climatological variables. More specifically, Creutin and

Obled (1982) and Tabios and Salas (1985) demonstrated that for monthly rainfall and storm totals it is preferable to other rainfall interpolation methods. Goovaerts (2000), Lloyd (2005), Basistha et al. (2008), Ly et al. (2011) confirmed these results.

Generally, Kriging can be applied to a wide range of datasets (e.g., Stahl et al., 2006; Phillips et al., 1992), and allows for the estimation of the variance of interpolated quantities (e.g., Li and Heap, 2011). The interpolation can be improved with the use of auxiliary variables, such as terrain-related parameters (e.g., relief, slope and aspect), as investigated in Attorre et al. (2007). Not surprisingly, Carrera-Hernández and Gaskin (2007) found that the use of elevation as a secondary variable improves temperature prediction.

However, Kriging can be computationally more demanding than other interpolation techniques. To overcome this problem, most applications that implement Kriging interpolators use either long time series with long time-steps, such as daily, (Verfaillie et al., 2006; Buytaert et al., 2006), monthly or yearly (Hevesi et al., 1992; Goovaerts, 2000; Boer et al., 2001; Todini, 2001), or short time series with shorter time steps (such as rainfall events) [e.g., Haberlandt (2007)]. Having tools that implement efficient computations could help to extend the interpolation method to real time processes.

Based on these premises, we set ourselves two objectives with this work. The first was to provide an efficient and precise tool for spatial estimations and interpolation of environmental quantities. The second was to make use of an implementing strategy that favors the usability of the software, its maintenance, its inspection, its extension and, hopefully, makes scientific work easier. This second goal comes under the contemporary efforts to promote open science (e.g., https://www.fosteropenscience.eu/). However, in order to maintain the right focus, we will not discuss the open science aspects and philosophy directly in this paper, rather we will present the Kriging software and its design.

The Spatial Interpolation Kriging package (version 0.9.8) (GEOframe-SIK, henceforth simply SIK) is here presented. It is a package that makes estimates of any spatially distributed environmental data at hourly steps (or sub-hourly when it is reasonable). SIK is designed according to the Object Modeling System v.3 (OMS3) framework (David et al., 2013), as such it is compatible with the GEOframe-NewAGE system, (Formetta et al., 2014; Bancheri, 2017). As a consequence, the package can be integrated with other GEOframe-NewAGE components and connected to them at run-time to form a variety of Modelling Solutions (MS). In this work, the SIK package is presented as four components:

1. the first is used for the production of the experimental semivariograms;

2. the second is used for the production of the theoretical semivariograms;

3. the third is used for the Kriging interpolation;

4. and the last is used for an automatic and easy jackknife resampling to asses the error of estimates.

SIK inherits some previous code used, for instance, in Formetta et al. (2014) and Abera et al. (2017). In particular Abera et al. (2017) assessed the effects of interpolation of precipitation on long-term mean annual runoff.

To make the code more flexible, easily extensible, and maintainable, SIK was completely refactored and a systematic use of Design Patterns (DP), (Gamma, 1994; Freeman et al., 2004), was introduced.

Several geostatistical tools have been made available to the scientific community. Among them, PyKrige (https://github. com/bsmurphy/PyKrige), SAGA GIS kriging (www.saga-gis.org), GRASS (grass.osgeo.org), Surfpack (https://dakota.sandia. gov/content/surfpack), R $gstat$ (www.cran.r-project.org) and the High Performance Geostatistics Library HPGL (https://www. github.com/hpgl). However, only some of these can be considered as alternatives to SIK, i.e. the ones that are open source,
comprehensively documented and actively developed:

- Dakota (Surfpack): C++ software with flexible interface that provides optimization algorithms, uncertainty quantification, parameter estimation, and sensitivity analysis for supporting computational models and simulators (Adams et al., 2009);

- PyKrige: python package that does 2D and 3D ordinary and universal Kriging computation with flexible design for
custom variogram implementation (Murphy, 2014);

- $gstat$: R package (computational core coded in C) that supports block Kriging, simple, ordinary and universal (co)Kriging and many other features (Pebesma, 2004), (Gräler et al., 2016). It is historically the leading software in this field.

While GIS-based tools, such as QGIS and GRASS (v.kriging) Krigings, are easily included into scripts leveraging GIS capabilities, they are not easily included into complicated MS.

As well as being open-source, SIK is the only Java-based and component-based software of those mentioned above. Moreover, it implements a quick way to plug-in to hydrological models and automatic calibration algorithms. We decided to compare the performances of SIK and R $gstat$, since the latter is one of the most widely used tools in the scientific community.

The present paper is organized as follows. First, some preliminary information on Kriging interpolation is given in section 2. Then the structure of the package and the informatics are presented in section 3. Section 4 describes the study area and
the experimental setup. The results of the application of the SIK package to temperature and rainfall datasets are discussed in section 5. Finally, a comparison of results with the interpolation of both datasets obtained with the R $gstat$ is presented in section 6.

## 2  Algorithms required for kriging

Kriging is a group of geostatistical techniques used to interpolate the value of random fields based on spatial autocorrelation of
measured data (Isaaks and Srivastava, 1989; Goovaerts, 1997; Kitanidis, 1997). The theory is briefly summarized in Appendix A.

Three main variants of Kriging can be distinguished, (Goovaerts, 1997):

- Simple Kriging (SK), which considers the mean to be known and constant throughout the study area;

- Ordinary Kriging (OK), which accounts for local fluctuations of the mean, limiting the stationarity to the local neighbor-
hood. In this case, the mean is unknown.

– Kriging with a trend model (here Detrended Kriging, DK), which considers that the local mean varies within the local neighborhood.

The trend can be, for example, a linear regression model between the investigated variables and an auxiliary variable, such as elevation or slope. According to the procedure shown in Goovaerts (1997), the DK is performed as follows: i) the trend is subtracted from the original data and the OK of the residuals performed, ii) the final interpolated values are the sum of the interpolated values and the previously estimated trend.

Variants of OK and DK are local ordinary kriging (LOK) and local detrended Kriging (LDK) respectively. In this case the estimate is only influenced by the measurements belonging to a neighborhood, which are usually defined either in a maximum searching radius or as a set number of stations which are closer to the interpolation point. In the LDK case, the trend is estimated locally too, and therefore it can take account for local trend variations.

The SIK package implements both the OK and the DK, since local mean may vary significantly over the study area and the SIK assumption about the mean could be too strict (Goovaerts, 1997).

The workflow of the main algorithm for solving an interpolation problem with Kriging can be summarized in the following steps:

1 - get the data from gauges;

2 - build the empirical semivariogram;

3 - fit a theoretical model to the semivariogram;

4 - use the theoretical model for solving the Kriging system;

5 - produce continuous surface maps or pointwise time-series of the quantity desired in any point of the domain;

6 - calculate estimation errors.

The last step underlines that we are not only interested in estimating a variable (temperature, rainfall intensity or other scalars) but also in evaluating the errors of our estimate. Besides the spatial variable estimate, Kriging also returns a variance of the estimate. However, Goovaerts (1997) states that the standard deviation cannot be used as a direct measures of estimation precision, since the Kriging variance is only a ranking index of data geometry (and size) and not a measure of the local spread of errors (Deutsch and Journel, 1992).

Therefore, to estimate the errors produced by Kriging interpolations, we chose the Leave-One-Out (LOO) cross validation technique (Efron, 1982; Isaaks and Srivastava, 1989; Martin and Simpson, 2003; Aidoo et al., 2015). LOO cross validation consists of removing one data point at a time and performing the interpolation for the location of the removed point by using the remaining stations. The approach is repeated until every sample has been, in turn, removed and estimates are calculated for each point. This procedure is straightforward, but cumbersome if performed manually. Therefore, a special module (component) was programmed and implemented to do it. LOO estimates errors just over the location where measures are available and, eventually, these errors can be interpolated themselves to obtain an error estimation in any point of the spatial domain.

## 3 Design, deployment and use cases of the SIK package

On the basis of the analysis of the mathematical problems and of the use cases delineated in the previous section, the design of the software was organized in four OMS3 components, the logic of which is explained below.

### 3.1 Overall design of the SIK components

The component-based environmental modeling framework OMS3, (David et al., 2013), was chosen for the development of the SIK code. Components are self-contained building blocks, modules or units of code (e.g., Argent, 2004; Van Ittersum et al., 2008). Each component implements a single modeling concept and the components can be joined together to obtain an MS that can accomplish a complicated task. The OMS user does not need extensive knowledge of OMS libraries. As its authors state in David et al. (2013): "there are no interfaces to implement, no classes to extend, no polymorphic methods to override

and no framework-specific data types to use". Besides, when the workflow allows it, components are run in parallel without any special effort by the computer programmer (this property is often called "implicit parallelism").

Besides minimizing couplings, the advantage of building within a modular software framework is the production of a code that is more flexible and easier to maintain and be inspected by third parties. Multiple algorithms can be implemented within the same component or in various components and inserted in the MS as alternatives. Thus, inside the same chain of tools,

different candidate solutions to the same hydrological problem can be compared. More details on OMS3 can be found in David et al. (2013), Formetta et al. (2014) and in Bancheri (2017). It is clear that the adoption of such a framework as the basis for our programs has an impact on the software design. However, the initial implementation of Kriging, used in Formetta et al. (2014) and in Abera et al. (2017) was designed to group all the tasks into a single component. In this case we thought it useful to split it into four components: the first is SIK-EV, related to the production of experimental variogram from data; the second is

SIK-TV, related to the selection and parameter estimation of the theoretical variogram; the third is SIK-K, for the solution and mapping of the Kriging system; and the fourth is SIK-LOO, to manage the error estimation. SIK-LOO does not work alone to produce its results, it uses the other three components to generate the spatial estimates of errors.

Figure 1 shows the MS for the interpolation and validation process in OMS. Each component is represented by a rectangle with rounded corners containing the name of the component itself and its inputs and outputs. Arrows represent the connections

between components. This representation is also used in the subsequent sections. The inputs for SIK-EV are the time series of the measured variables and the geometry of the measuring stations, in shapefile format with the spatial coordinates. The outputs are the experimental variogram values and the distance vector.

The distances vector, the name and the parameters for the theoretical semivariogram models (sill, nugget and range), are the inputs of SIK-TV. Particle Swarm Optimization (PSO), (Eberhart and Kennedy, 1995), is the component that optimizes

the theoretical model parameters. Further inputs for the calibrator are the objective function to be optimized and other internal parameters, such as the number of iterations and the tolerance. PSO can be connected to the SIK-K component (blue dashed arrow) or to the SIK-LOO component (red dashed arrow). Inputs for SIK-K are the shapefiles with the coordinates of the measuring stations and the interpolation points, the measurement data, the DEM and the optimized parameters of the semivar-

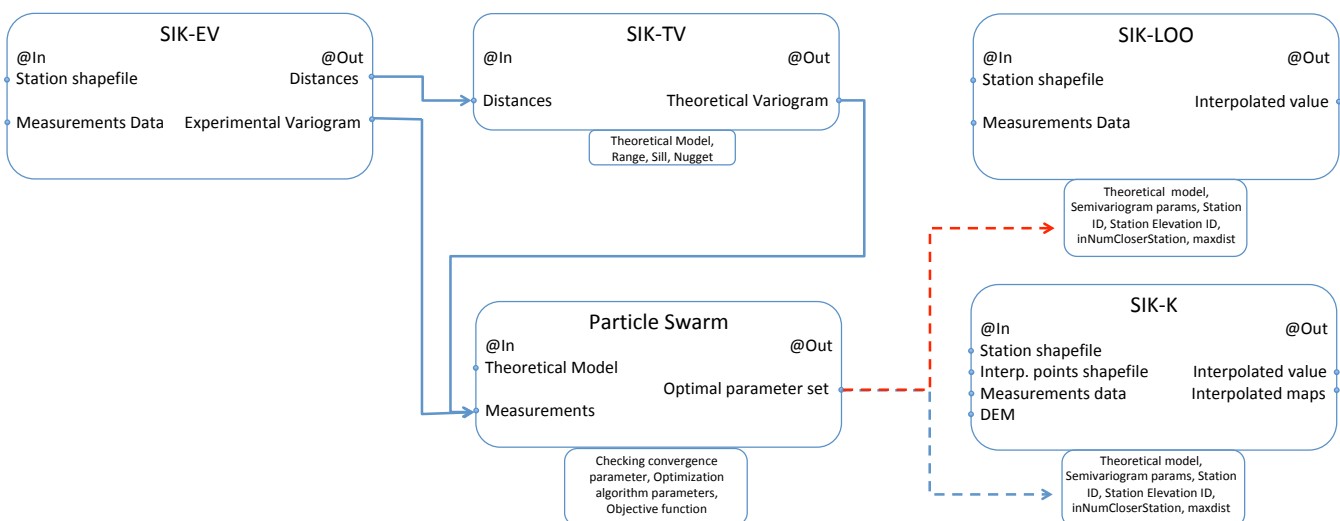

**Figure 1.** Flow chart of interpolation and validation process represented with the relative OMS components.

iogram model. Final outputs are either time series or maps of interpolated values, which can be visualized directly in a GIS system.

Using the alternative connection, PSO can be connected to SIK-LOO, which implements the iterative procedure necessary to estimate errors of interpolation. Given $n$ spatially distributed measures, $n-1$ measures are used for the interpolation, while the remaining one is used for comparison to produce the error estimate. The operation is repeated $n$ times, each time excluding a different gauged location, to obtain a set of $n$ estimated errors. Because our package needs to deal with time-varying fields, the operation is repeated for each time step, when measures are available, and the site error is actually a temporal mean over a period.

## 3.2  Internal classes design characteristics

Each of the four OMS3 components presented in the previous section can contain alternative solutions. For example, in the SIK-TV the software design allows for multiple theoretical variogram models, while in the SIK-K component the four types of Kriging listed in section 2 were implemented.

In principle, we could have implemented a single component for every single type of variogram and single type of Kriging but this should have exploded the number of software modules to maintain. However, to "close the code to modification and keep it open to extensions" (Martin, 2002) and to maintain the code abstract enough to avoid code disruption at any addition ("program to an interfaces", e.g., Gamma 1994), we adopted the use of DP (Gamma, 1994). This was a further enhancement with respect to the previous version of the SIK package.

In general, DP implement rules that allow, for instance, to separate code parts that are going to vary from those that are going to remain the same. The adoption of these DP, once their rationale is understood, makes the code easier to be read

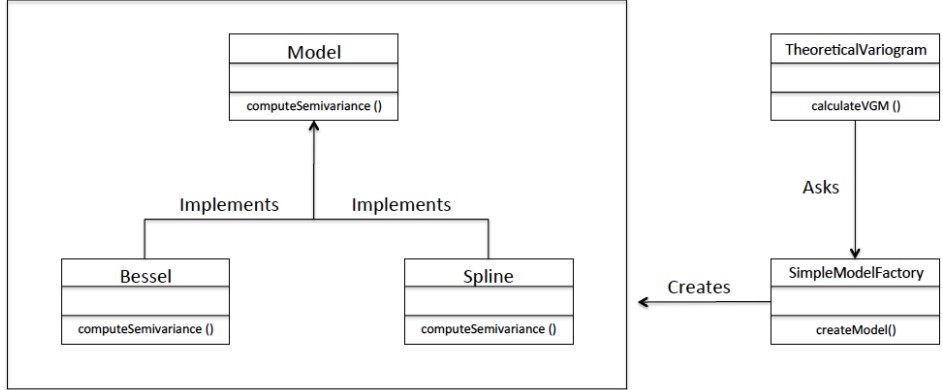

**Figure 2.** Implementation of the Java simple factory for the choice of the theoretical variogram model in the component SIK-TV.

and maintained. While largely known among programmers, DP did not penetrate much in the scientific community, which has remained largely impervious to these techniques, and just a few examples of good practice can be found in the scientific literature (Gardner and Manduchi 2002; Donatelli and Rizzoli 2008; Rouson et al. 2011).

The various theoretical semivariogram models or Kriging types to be chosen at run-time were encapsulated by using the
Simple Factory class (Freeman et al., 2004). In this way, adding a new type of variogram to, or deleting an obsolete one from, the SIK-TV is straightforward and requires few changes, which are confined to just a class.

Figure 2 shows the implementation of the Simple factory class for the choice of the theoretical semivariogram model. The concrete classes, Bessel and Spline, implement the same interface, Model. The Simple factory (named SimpleModelFactory in the Figure) generates objects of a concrete class from given information (i.e. a string containing the name of the chosen model).
The component SIK-TV (named TheoreticalVariogram in the Figure) class uses the pattern to get the object of the concrete class.

The dependency inversion principle, according to which high-level modules shouldn't depend on low-level modules, (Eckel, 2003), was also strictly respected in all the programming. The dependencies of the classes are not demanded by the concrete subclasses but only by the abstract classes and interfaces, (Ellis et al., 2007). So any changes in a concrete (sub)class do not
affect the overall structure of the program and remain limited to it.

### 3.3   Use cases

Figure 3 exemplifies how to plug-in the SIK package to the hydrological model GEOframe-NewAGE. The MS presented allows one to estimate the relevant variables of the hydrological budget: after the spatial interpolation of precipitation, it is possible to simulate the components of the energy budget, shortwave and longwave radiations, the snow processes of accumulation and
melting, and the potential evapotranspiration. These are then used as inputs for the rainfall-runoff model, Embedded Reservoirs model, (Bancheri, 2017), which provides discharge and actual evapotranspiration. Thanks to the MS, it is possible, for example,

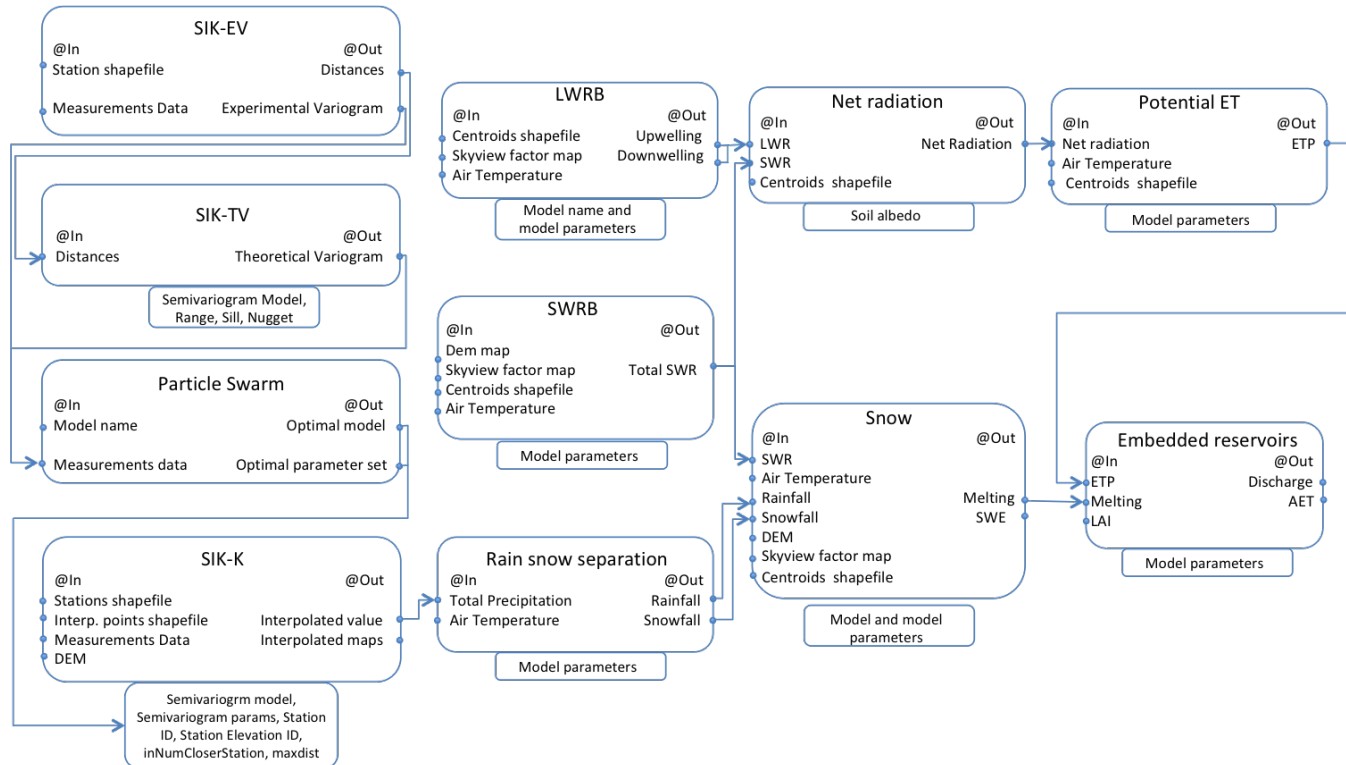

**Figure 3.** Flow chart of the connection of SIK to a GEOframe-NewAGE configuration as used in Bancheri (2017). It allows the determination of the main elements of the water budget of a catchment.

to test the impact of the type of Kriging used on the rainfall-runoff model outputs or to perform a validation of the SIK package using remotely sensed data, e.g. MODIS data product, (Turk and Miller, 2005; Hall et al., 2006; Abera et al., 2017). Further information about this MS solution are presented in Bancheri (2017).

A second MS is presented in Figure 4. This MS interpolates temperature maps, which are the inputs for the shortwave and
5   longwave radiation components and for the Shymansky-Or EvapoTranspiration (SO-ET) component, built after (Schymanski and Or, 2017). Inputs of the SO-ET component are the maps of interpolated temperature, the shortwave and longwave radiation maps and DEM. Outputs are the ET maps and leaf temperature. Obviously, it is possible to use SO-ET instead of Potential ET in Figure 3. In that case, two sets of Kriging act concurrently to give maps of temperature and rainfall. Another different scheme (not shown) is obtained when the parameters of the radiation decomposition model, (Formetta et al., 2013, 2016), i.e.
10   those parameters which are used to determine the attenuation of radiation due to the atmosphere, are set to be spatially varying. In this case a further (third) Kriging set is run.

Figures 3 and 4 are just two examples of MS, point-wise and raster, that can be obtained by plugging-in the SIK package to the other components available in GEOframe-NewAGE. The flexibility of the package allows one also to use it as a stand alone, opening up the possibility of using it with other MS involving other softwares.

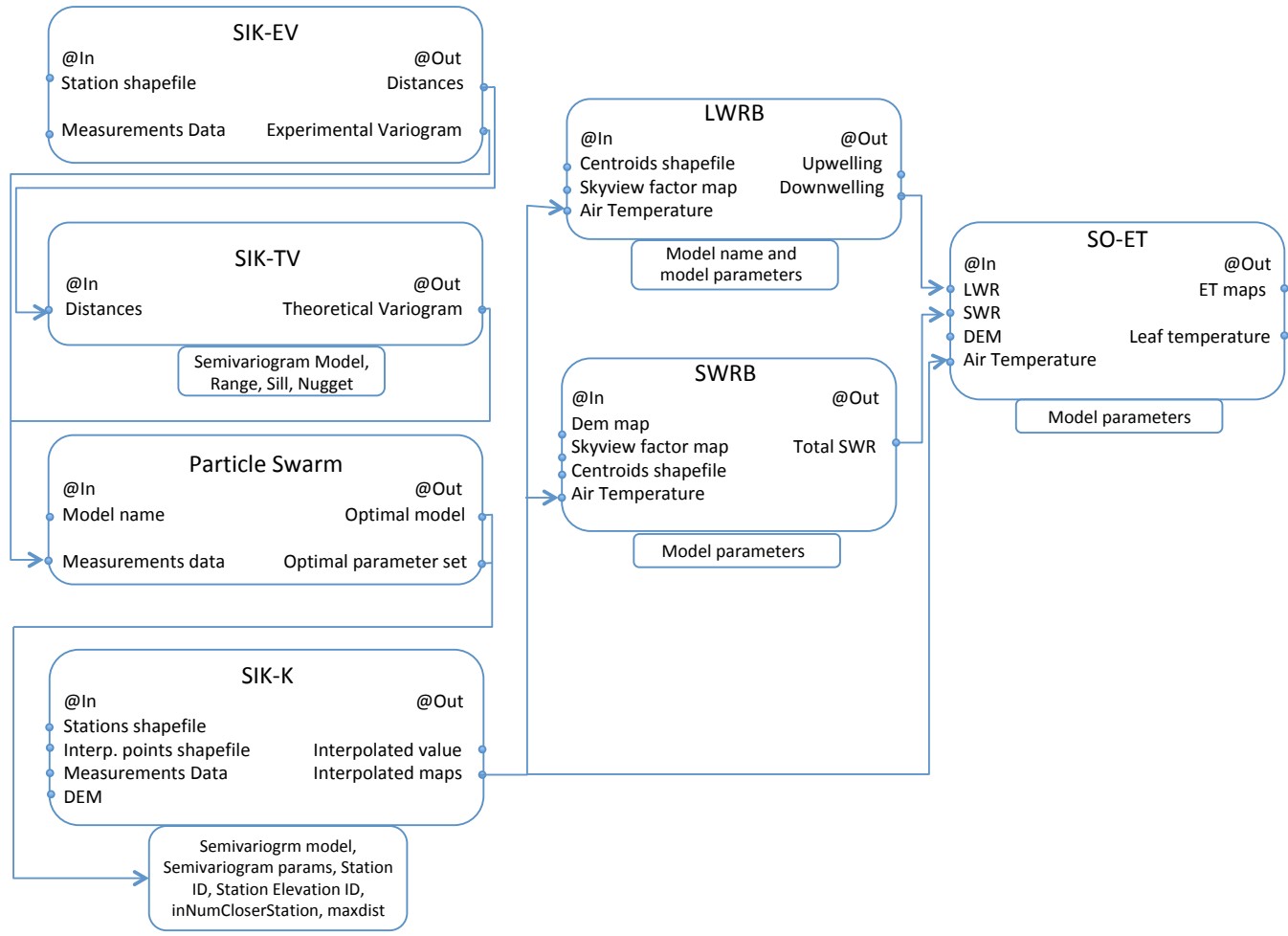

**Figure 4.** Flow chart of the connection of SIK to the SO-ET component. In this MS, the maps of the temperature, input of SO-ET, are interpolated.

## 3.4 SIK development as an RRS tool

Here, we delineate the practices implemented in building the SIK package for making it a Reproducible Research System (RRS), (e.g. Formetta et al. (2014)).

Although the initial code (let's call it v 0.1) was already available from a control version system under GPL v3 license (www.gnu.org/licenses/gpl-3.0.en.html), the repository was owned by the original author. A non-personal repository was judged to be better suited to host a collaborative work. Therefore, for SIK and its companion tools the collective GEOframe organization repository was created under Github (www.github.com), using Git (www.git-scm.com), and can be found at the following link: www.github.com/geoframecomponents.

Code v 0.1 did not include a building tool. These tools can be considered a modern evolution of the Unix "make" (e.g., www.gnu.org/software/make/) and take care of gathering the various concurring libraries and linking them to form the final executable file. In our case, the possible choices for Java projects include Apache Ant (www.ant.apache.org), Maven (www.maven.apache.org) and Gradle (www.gradle.org). All of these provide ways to solve the software dependencies. Both Maven and Gradle can download and update the remote resources needed. Our final choice was Gradle, since it uses a more concise syntax, thanks to the use of the Groovy language (www.groovy-lang.org), compared to the XML (www.w3.org/XML) used by Maven. Using building tools also allows at abstracting from the use of IDEs. In the current Java market there are at least three major IDEs for managing large projects: Netbeans (www.netbeans.org), Eclipse (www.eclipse.org) and IntelliJ (www.jetbrains.com). All of them support both Gradle and Maven and Ant and can import a Gradle or Maven (or Ant) project seamlessly. These tools are widely used by programmers, but rarely by scientists, who are increasingly struggling with the difficulties of maintaining their own code. With these tools, researchers could master others' codes more easily, especially if they are open source. Therefore, we think that adopting a proper building tool is useful in promoting collaborative work and open science.

Another important step in the management of the code was the implementation of a continuous integration system (http://www.jenkins.io). It ensures the building and testing of the source code at each commit, forcing the good practice to prepare tests for each software module developed.

Continuous integration (Meyer, 2014), is the practice of merging all developer working copies to a shared mainline several times a day. Unit Tests (Beck, 2003) are built with the code and run each time the merging is done. The continuous integration service automatically builds the executable codes, checks if the tests are performed correctly and returns a positive answer if all is done properly. Eventually, major code commits are tagged with release numbers, under the GPL v3 license. For this purposes, we chose to use Travis CI (https://travis-ci.org), which uses GitHub as a web-based git repository hosting service, and is a good choice for a continuous integration service.

Since Github is a repository and not an archival system, we decided to use Zenodo (www.zenodo.org) to provide our products with a Digital Object Identifier (DOI) and then we put the entire project, as used to obtain the results presented in this work, on Open Science Framework (www.osf.io). The assignment of the DOI allows researcher peers to retrieve exactly that code in the foreseeable future. This could be important when reconstructing which software version was used in a paper and, perhaps, it could make life easier within research groups.

## 4 Testing and simulations setup

### 4.1 Study area and data description

To test the performances of the modeling solutions presented in Figures 3 and 1, we used the SIK components to interpolate temperature and rainfall data from 97 stations located in the Isarco River valley, Italy, shown in Figure 5 and detailed in the complementary material. The Isarco River is a left tributary of the Adige River, in the Trentino-Alto Adige Region, Northern Italy.

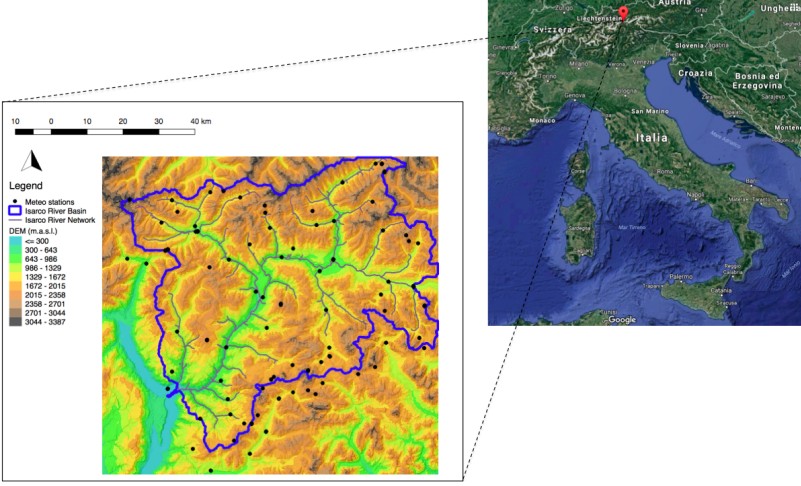

**Figure 5.** Geo-location of study area and position of meteorological stations.

The catchment area is about 4,200 km$^2$ and the altitude ranges from 210 m a.s.l. to 3,400 m a.s.l. The river length is about 95 km and the discharge is about 78 m$^3$/s yearly average at its confluence with Adige River near Bolzano. The geological and geomorphological conditions of the valley are homogeneous and represented by the very thick Palaeozoic series of the "Complesso Vulcanico Atesino". The formation is deeply affected by the effects of Quaternary glacial and post-glacial erosion. The cross-section of the Isarco Valley along the Bolzano-Ponte Gardena stretch is characterized by very steep and rugged slopes. Climate is typically alpine, characterized by dry winters, snow and glacier-melt in spring, and humid summers and autumns. The land is mainly used for agriculture in the upper part, while in the lower part of the basin, the narrow valley is mainly occupied by civil infrastructures.

Data used for the testing were provided by Provincia Autonoma di Bolzano (local government), and collected into the Adige database (http://abouthydrology.blogspot.it/2016/09/the-adige-database-or-database-newage.html) during the CLIMAWARE and GLOBAQUA projects. The Digital Elevation Model (DEM) of the study area was downloaded from the USGS EarthExplorer (https://earthexplorer.usgs.gov) and it has 100 m x 100 m cell size.

### 4.2 Simulation setup

In the available dataset (2003-2013) we identified the year with the smallest number of missing data, which was the 2008, and then we used it to test the SIK components.

A quality check was made to eliminate any outliers. Also, the spatial distribution of the no-value was analyzed, in order to assess the number of bins of distances in which to compute the semivariance. In fact, to reduce the number of points in the experimental semivariogram, the pairs of locations are grouped based on their distance from one another. This grouping

process is known as binning. For each time step, we found that about 10% of stations were not recording data. Therefore, since the mean number of active stations for each time step was 70-80, we decided to use 8 bins. This choice was also supported by a visual inspection of the shape of the experimental semivariance, which confirmed that by using 8 bins the number of stations involved were neither too low nor too high.

In order to assess the goodness of SIK performances, two applications were performed:

- an interpolation of one year of hourly temperature data;

- an interpolation of a rainfall event, also at hourly time steps.

Firstly, the analysis of the semivariance was performed and experimental semivariograms were fitted using all the 11 theoretical models. The model that gave the best fitting was then used for the interpolation of the temperature and rainfall variables

using the 4 types of Kriging. Kriging performances were assessed using the leave-one-out cross validation. The two local cases (LOK and LDK) were performed using a fixed number of closer stations. In particular, we decided to use 10 stations for the temperature case, since it was a good compromise between the distance among the stations and the mean number of recording stations for each time step. Regarding the local interpolation of precipitation, the number of closer stations was 5, given the prevalently convective nature of summer precipitation and the lower number of active gauge stations for each time step. Finally,

results obtained from the interpolation of the temperature dataset were compared to the results obtained with R $gstat$, in order to assess the differences between the two packages, their easiness of use and their performances.

## 5    Simulations results

### 5.1    Application of SIK to a temperature dataset

The first application of SIK components was done using the temperature dataset. The hourly experimental semivariograms

were computed and then fitted using the 11 available theoretical models.

Figure 6 shows the results of the fitting of the experimental semivariogram for a single time step on 15 June 2008. The black dots represent the experimental semivariance, while each colored curve represents a different optimized theoretical model. The Y and X axis show the values of the semivariance $\gamma$ [h] and the distances in meters respectively.

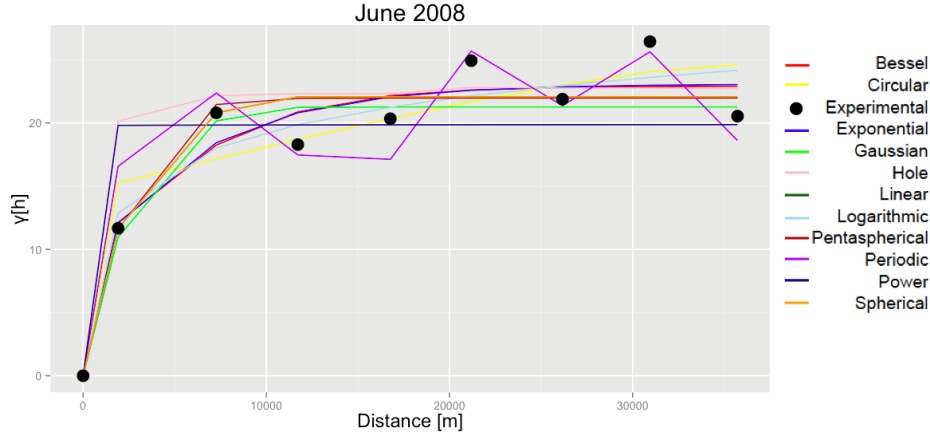

**Figure 6.** Fitting of the experimental semivariogram using PSO for 15 June 2008 12:00 CET.

Table 1 reports the main indexes of goodness of fit (GOFs), namely NSE, RMSE, $R^2$ and PBIAS, computed between the experimental semivariogram and the 11 theoretical semivariogram models. The aforementioned GOFs are defined in Appendix C and were obtained using the R package $hydroGOF$ (https://cran.r-project.org/web/packages/hydroGOF/hydroGOF.pdf), which computes the GOFs between measured and simulated values (in this case experimental semivariance and theoretical semivariance). All the semivariogram models gave satisfactory results, with large values of NSE (0.72 : 0.92) and $R^2$ (0.73 : 0.92), low values of RMSE (2.14 $^oC$ : 3.99 $^oC$) and PBIAS (-3.80% : -7.90%), confirming the accuracy of the calibration procedure.

| Semivariogram | NSE | RMSE | $R^2$ | PBIAS |
|---|---|---|---|---|
| Bessel | 0.92 | 2.14 | 0.92 | -0.20 |
| Circular | 0.88 | 2.59 | 0.88 | 0.0 |
| Exponential | 0.92 | 2.10 | 0.92 | -3.80 |
| Gaussian | 0.90 | 2.39 | 0.91 | 0.35 |
| Hole | 0.77 | 3.61 | 0.81 | 7.90 |
| Linear | 0.91 | 2.28 | 0.91 | 0.0 |
| Logarithmic | 0.92 | 2.17 | 0.92 | 0.0 |
| Pentaspherical | 0.91 | 2.29 | 0.91 | 0.0 |
| Periodic | 0.90 | 2.18 | 0.92 | 0.0 |
| Power | 0.72 | 3.99 | 0.73 | -3.70 |
| Spherical | 0.91 | 2.28 | 0.91 | 0.0 |

**Table 1.** Performance results of semivariogram models used.

In order to asses the goodness of the interpolation, we performed the leave-one-out cross validation using the optimized hourly values of sill, nugget and range for the Bessel model, which is one of the best semivariograms according to the previous results.

Figure 7 shows the results for the four types of Kriging in terms of NSE. Each point represents the averaged monthly NSE over the 97 meteorological stations. The two local cases were performed using the ten closest stations to the interpolation point. For both the OK and LOK cases the performances were very poor (NSE < 0.5), indicating that mean temperature might have been a better predictor than the interpolation.

A strong trend between temperature and elevation ($R^2 \sim 0.9$) was detected during the quality check phase (which was expected). Therefore, interpolation results obtained using the DK and the LDK present optimal higher values of the goodness of fit index (maximum NSE of 0.93) compared to the OK and LOK cases.

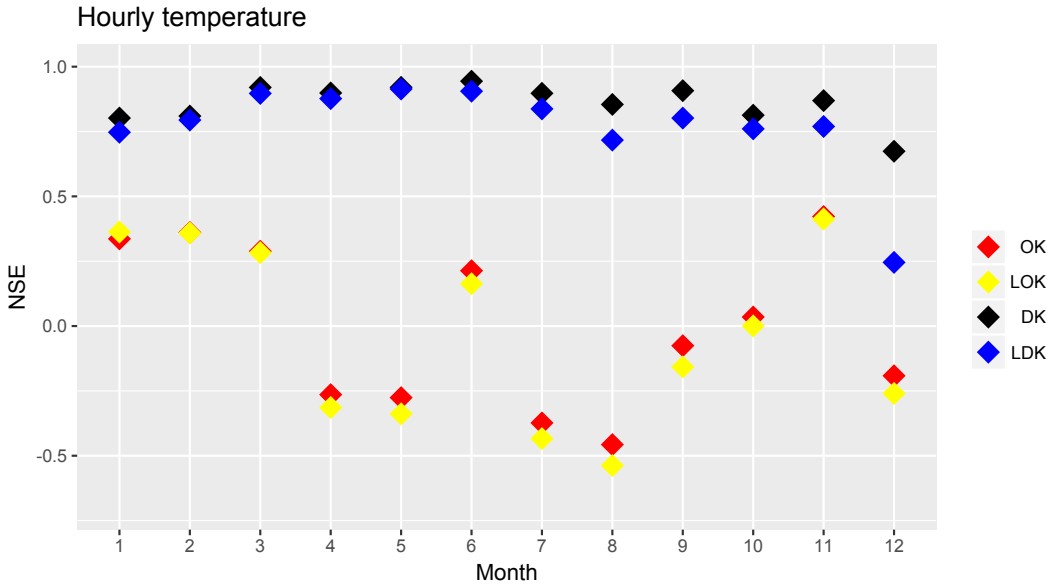

**Figure 7.** Monthly variation of the NSE index over the entire hourly temperature dataset using the Bessel semivariogram model.

The spatialization of temperature was done for each pixel of the DEM, applying the LDK and the Bessel semivariogram model. Figure 8 shows the maps obtained for two different dates in the 2008, one in winter (15 February 2008 12:00) and one in summer (15 June 2008 12:00). The bubble plots of the RMSE obtained between the measured and the interpolated values have been overlapped onto the maps. The size of a bubble represents the magnitude of the error: the largest error for the interpolation of February 2008 (RMSE = 4.1 $^o$C) corresponds to Station ID 90534 (Z=1385 m a.s.l.), while the largest error for June 2008 interpolation (RMSE = 3.2 $^o$C) corresponds to Station ID 90266 (Z=490 m a.s.l.).

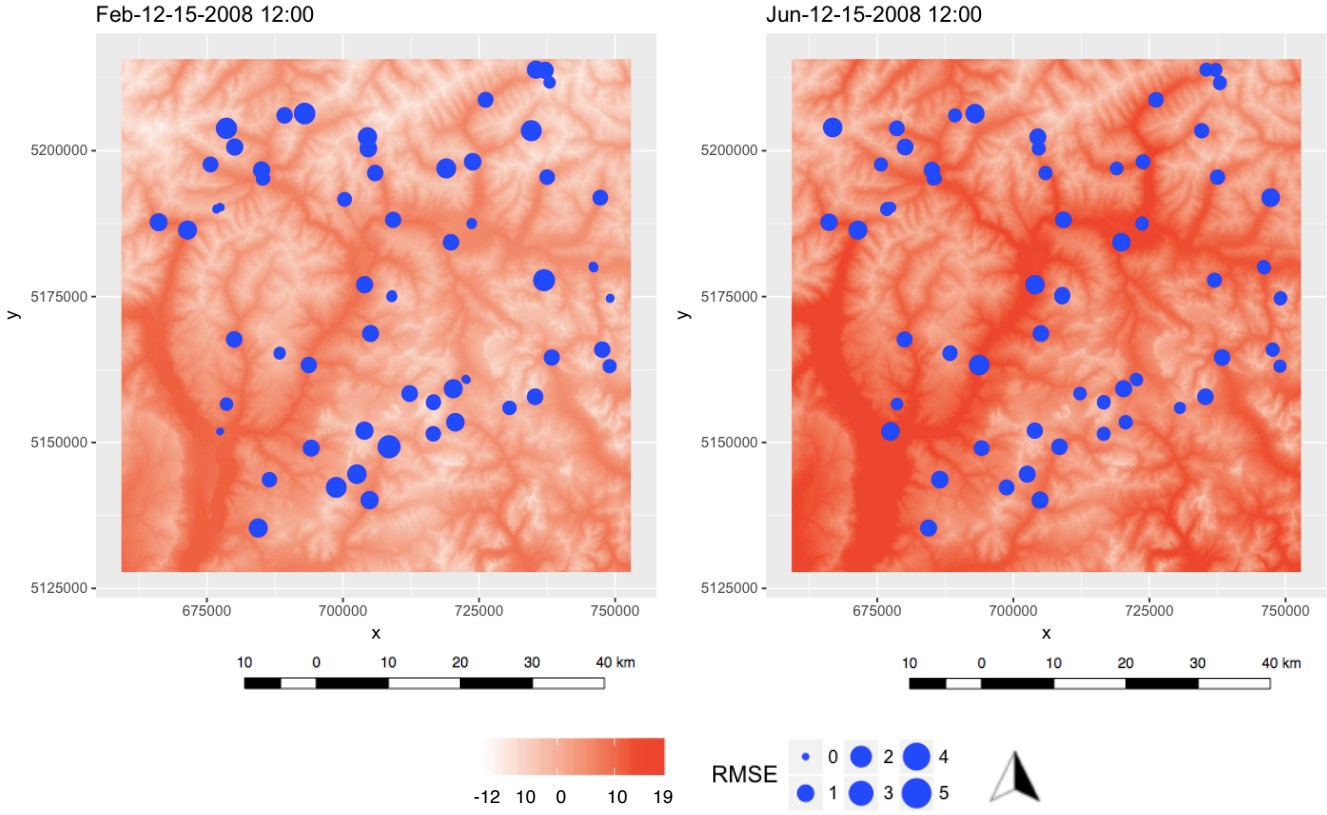

**Figure 8.** Maps of spatialized temperature for 15 February 2008 and 15 June 2008. Two bubble plots are overlapped, which represent the RMSE between the measured and interpolated values.

## 5.2 Application of SIK to a rainfall dataset

The application to a rainfall dataset was made at event scale; specifically, a rainfall event of 11 h between the 29 and 30 June 2008. The event was chosen because it was the longest and most intense recorded by the highest number of stations for 2008.

Figure 9 shows the boxplots of the 11 hourly semivariograms with 8 bins of lag distance, while the red line represents the best theoretical semivariogram, which in this case was obtained using the Bessel model. The Y and X axis show the values of the semivariance $\gamma$ [h] and the distances in [m], respectively.

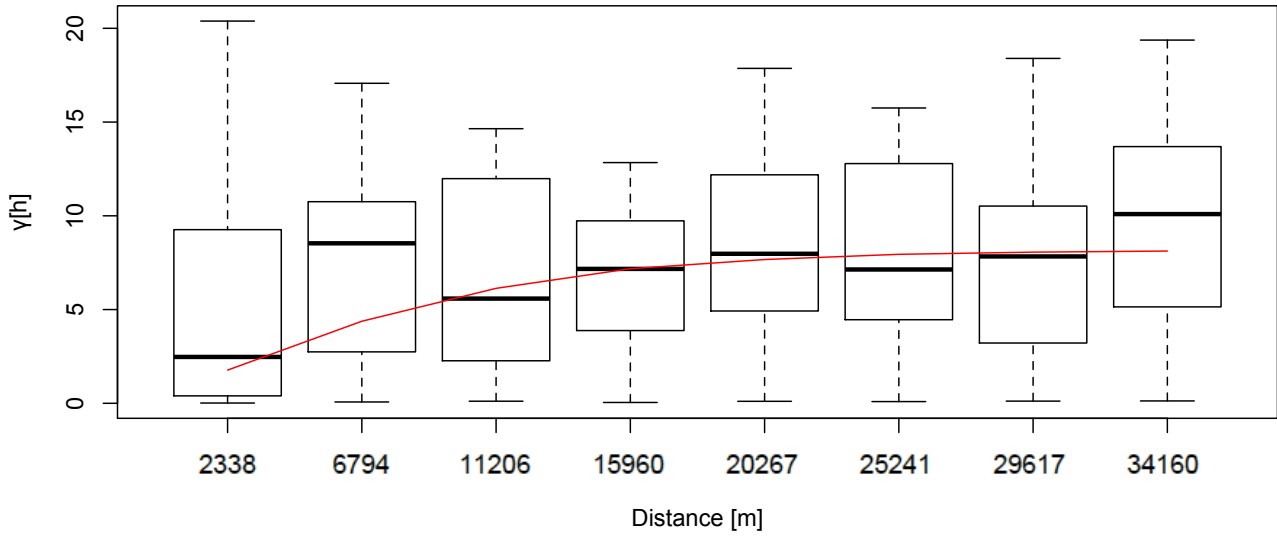

**Figure 9.** Boxplots of the semivariograms of the precipitation event of 29th and 30th June 2008.

The optimized values of range, nugget and sill were then used for the 4 types of Kriging interpolations. Figure 10 compares the results obtained for two stations (ID 1152 and ID 1270) chosen at different elevations (943 m a.s.l. and 2100 m a.s.l., respectively). All the interpolators were able to capture the rainfall peak at midnight for both stations. Comparing the volumes of cumulative rainfall, shown in Table 2, all the interpolators gave good results for the station ID 1152, while there is an overestimation in the case of station ID 1270; this is due to peaks detected but not recorded between the 21:00 and the 00:00.

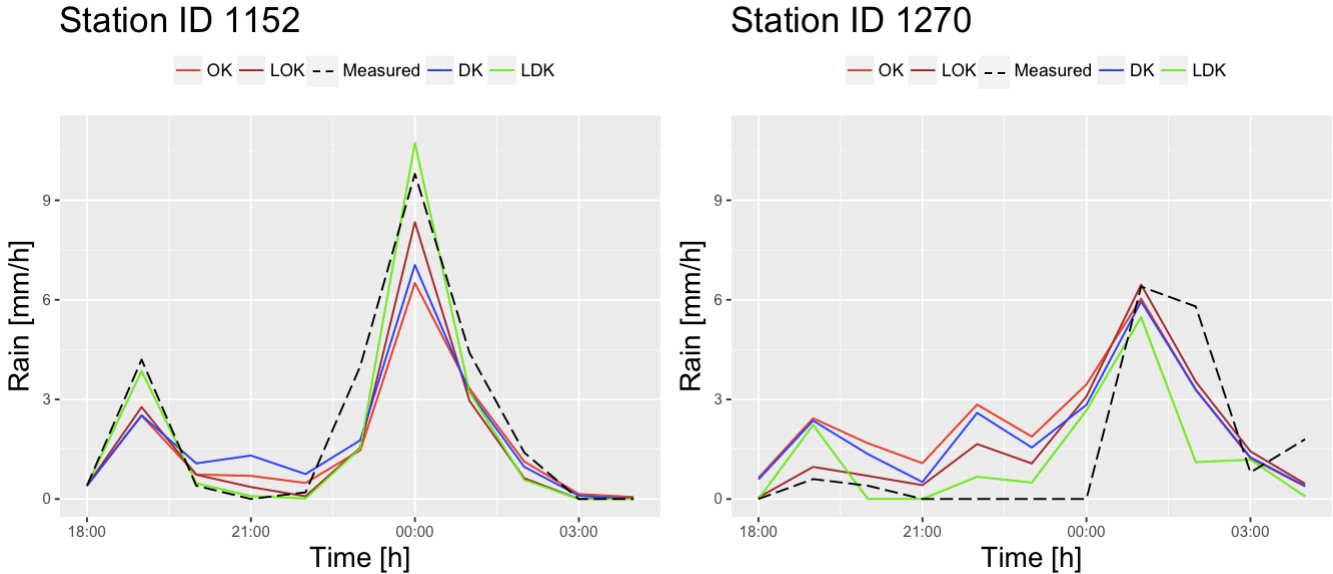

**Figure 10.** Comparison between the four types of Kriging and the measured rainfall.

Table 2 also shows the indexes of goodness of fit between the measured and the interpolated rainfall for the 4 types of Kriging and the two stations. The performances are overall good in the case of station ID 1152 and the best interpolator is the LDK computed using the 5 closest stations. Results for station ID 1270 are generally worse, with the highest NSE of 0.62 for the LOK case. This could be due to the higher elevation of the station, which led to a series of interpolated peaks that were not recorded.

| Kriging | ID 1152 | | | | ID 1270 | | | |
|---|---|---|---|---|---|---|---|---|
| | NSE | RMSE | $R^2$ | Cum (mm) | NSE | RMSE | $R^2$ | Cum (mm) |
| OK | 0.77 | 1.42 | 0.94 | 20 | 0.31 | 1.88 | 0.45 | 24 |
| LOK | 0.86 | 1.09 | 0.94 | 22 | 0.62 | 1.39 | 0.65 | 22 |
| DK | 0.80 | 1.33 | 0.83 | 22 | 0.46 | 1.66 | 0.54 | 20 |
| LDK | 0.91 | 0.92 | 0.92 | 25 | 0.35 | 1.82 | 0.37 | 13 |
| Measured | - | - | - | 25 | - | - | - | 16 |

**Table 2.** Results in terms of goodness of fit indexes between the measured and interpolated rainfall values for two stations.

The spatial interpolation of the precipitation was also done for each pixel of the DEM, applying the LOK and the Bessel semivariogram model. Figure 11 shows the results of the interpolation for 30 June 2008 at 00:00. As it appears from the map, the rainfall intensities are higher in the river valley, with a value of 9.8 mm/h measured at station ID 1152. The bubble plots

of the RMSE obtained between the measured and the interpolated values are overlapped. The size of a bubble represents the magnitude of the error: the largest error is obtained for station ID 90133 (Z=1246 m a.s.l.).

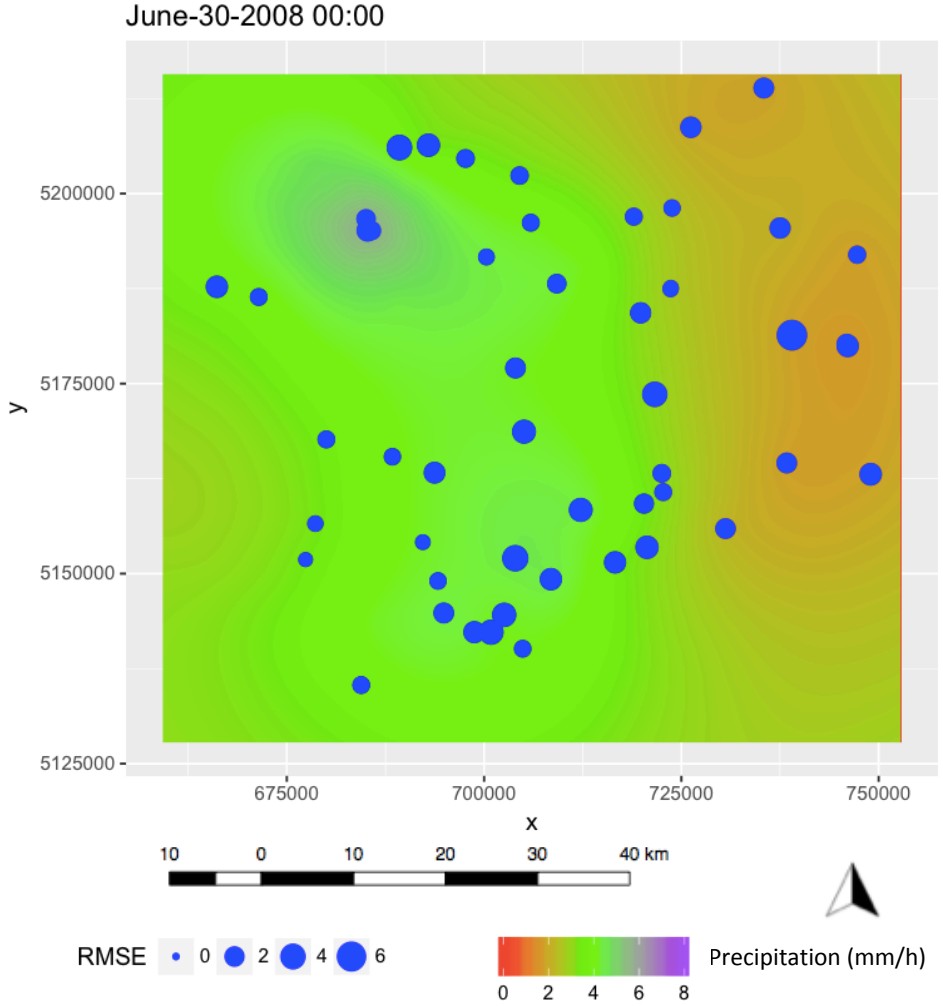

**Figure 11.** Spatial interpolation of the precipitation applying LOK and the Bessel semivariogram model. The bubble plot of the RMSE is overlapped.

## 6   A qualitative comparison with the R package $gstat$

"A comparison between SIK and the R package $gstat$ was made in order to highlight their differences and similarities, and to justify the deployment of an alternative software. We performed a qualitative comparison between the two softwares accounting for design, the implemented features, and the accuracy of the results. Benchmarks or quantitative performance comparisons

would not have been useful or completely truthful since the "velocity" of computation (a classic quantitative comparison) depends on too many factors, some of which are described below. Moreover, in our opinion, the two tools that we analyzed have different purposes. This can be seen just by looking at the features of the relative programming languages. The $gstat$ software is developed in C with a part of the code in R language. It must be executed using the various R environments. SIK is developed in Java (7) as a group of OMS components and it can be executed within the OMS console, as a stand-alone Java programs, or embedded in other codes in languages that support Java bindings. Java is slower than 3rd generation languages such as C. However, in the course of Java development several optimizations, such as "just-in-time compilation" and "adaptive optimization", have been introduced to improve the performance of its Java Virtual Machine (JVM). These techniques identify recurrently executed algorithms, so called "hot spots", and dynamically recompile them at run-time. Eventually, the hot spots gain valuable computational speed. C is one of the fastest compiled languages. But only the computational core of $gstat$ is coded in C; the management of temporal steps, such as "for-loops", and data structures must be scripted in R. Undoubtedly, R is a very powerful programming language, mainly because of its flat learning curve, and easy syntax and semantics, but it is fully interpreted, which makes it very slow. As a result, the comparison of the speed of computation for a single temporal Kriging interpolation is unfair against Java, since the JVM cannot exploit its optimization tools for a single computation. On the other hand, the comparison of the speed of computation for a year of hourly Kringing interpolations is biased against R, because temporal steps affect most of the computational time. In terms of functionality, $gstat$ computes both omnidirectional and directional semivariongrams, while SIK does not implement directional semivariograms yet (although we have included this feature on the software wish list). Furthermore, $gstat$ provides four more theoretical semivariogram models with respect to SIK: Matern, Matern with Stein's parameterizations, Wave, and Legendre. Adding the desired theoretical model to any SIK-TV component would be easy and straightforward, thanks to the DP implemented, as shown in Figure 2, but they are not available yet. Regarding the estimates that the two packages offer, these are usually different. Comparisons were made with both the temperature and rainfall datasets used in section 4. Semivariograms were computed using the same number of bins and cutoff distance.

Figure 12 shows the results of the temperature interpolations done with SIK and $gstat$, in terms of NSE, RMSE, PBIAS and $R^2$: the overall performances of both tools are very good. The NSE values are always above the 0.65, while the RMSE are always lower then 2 $^oC$.

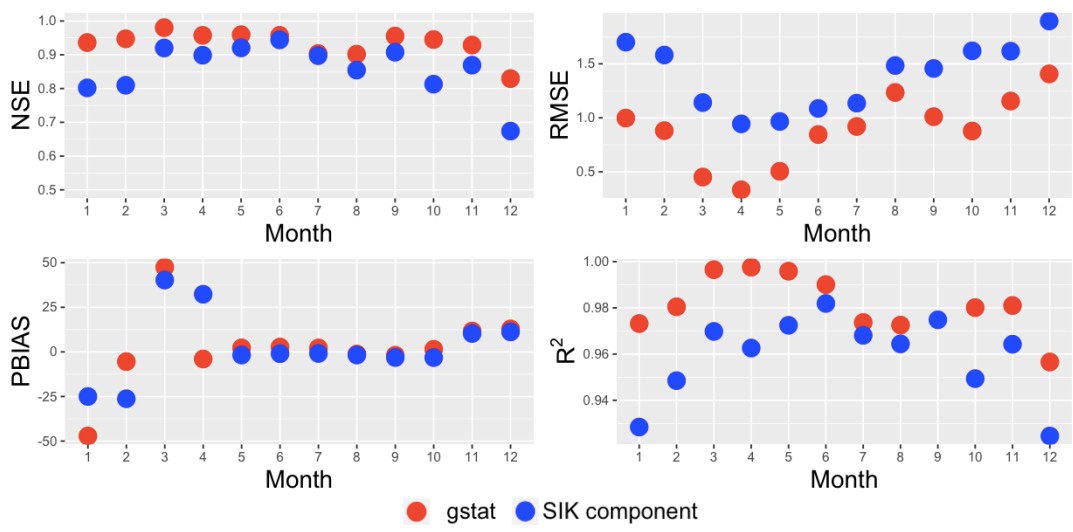

**Figure 12.** Comparison between the performances of $gstat$ and SIK packages in the interpolation of the temperature dataset.

Figure 13 shows the results of the precipitation interpolation done with SIK and $gstat$, in terms of NSE, RMSE, PBIAS, $R^2$ and cumulative volumes. Also in this case, both softwares are able to reproduce the rainfall event well, simulating the peaks. The results obtained for station ID 1152 are very good for both softwares, with a NSE >0.9. Both softwares show slightly worse results for station ID 2170, with lower values of NSE and $R^2$, higher RMSE, and an overestimation of the total rainfall (19 mm with $gstat$ and 20 mm with SIK, compared to the 16 mm recorded by the gauges).

In conclusion, *gstat* is a powerful, flexible tool to get fast results with fast scripting in answer to single, specific questions (with some implementation efforts user-side); SIK is a tool that is ready to be integrated into broader MS, specifically because of its OMS-compliant design. The interpolations of both temperature and rainfall confirm the quality and accuracy of the predictions obtained using the SIK package, demonstrating that it is a good competitor of R $gstat$.

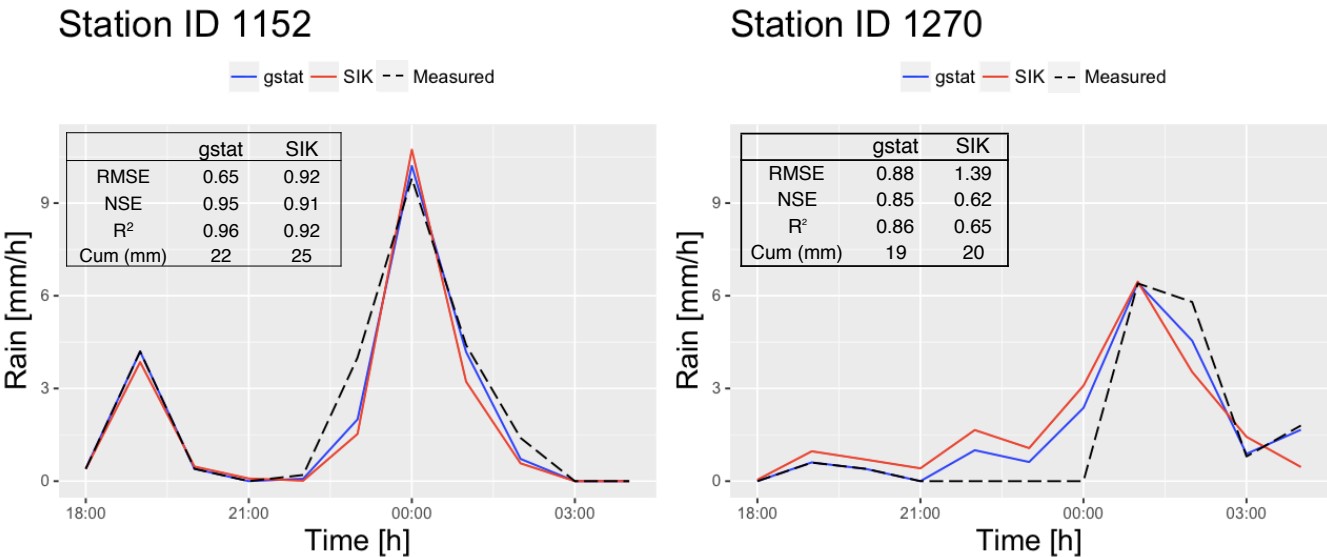

**Figure 13.** Comparison between the performances of *gstat* and SIK packages in the interpolation of the rainfall dataset.

## 7 Conclusions

This paper presents a new modelling package for the spatial interpolation of environmental variables. It includes 11 theoretical semivariogram models and 4 types of Kriging interpolations. To test the performance of the SIK package, two applications were performed: the interpolation of one year of temperatures and the interpolation of a rainfall event. Data were retrieved from a dataset of 97 stations located in the Isarco Valley in Italy and the resolution of the interpolation grid data was 100 m.

Several characteristics make the SIK package a good competitor tool among those available in the literature. From the user perspective:

- it can be used as a stand-alone;

- it can plugged-in to the hydrological modeling system GEOframe-NewAGE;

- it can be used with all OMS compliant components, such as calibration tools for the optimization of the parameters;

- it includes a tool for the automatic estimation of errors;

- its results are presented in data formats that can be visualized directly by GIS;

- a variety of MS can be obtained, according to user needs;

- it is faster than *gstat* in every-day use routine, under certain conditions.

From the programmer perspective, the implementation of DP makes the package easy to maintain and suitable for future improvements. All the elements are close to modification and open to extension. Further developments of the package are easy

and straightforward. Examples of such developments might include: integrating new types of Kriging, implementing a different selection method of the gauge stations, and the addition of non-linear relationships between the interpolated variable and an auxiliary variable.

The interpolations of both the temperature and the rainfall gave very good results, with a high agreement between the measured and the interpolated variables. The tests also show how it is possible to choose between 11 variograms and four Kriging alternatives and to compare the outcomes easily. On the other hand, the single rainfall event did not show trend with elevation.

In comparison with $gstat$, the SIK package proved to be a good alternative, regarding both the easiness of use and the accuracy of the interpolation.

**Computer Code Availability**

An OSF project with all the components needed to reproduce the results shown in this paper has been created and is available at the following link: https://osf.io/24rgv. The interested researcher can find the entire OMS project, containing input data, output, .sim files, jar files and R script used for the plots at the following link https://doi.org/10.5281/zenodo.1244034, besides in the OSF project. Moreover, the links to the source codes and to the documentation of the SIK components are also available in the OSF project. In particular, for the present work, version 0.9.8 is the version of the codes of the GEOframe-SIK package that we used, available at the following link: https://github.com/geoframecomponents/Krigings/tree/v0.9.8.

**Author contribution**

Marialaura Bancheri, Giuseppe Formetta and Francesco Serafin developed the model code integrated in the GEOframe-SIK package. Marialaura Bancheri, Francesco Serafin, Michele Bottazzi and Wuletawu Abera designed the experiments and performed the simulations. Riccardo Rigon planned the research, coordinated and supervised all its phases, and provided the financial support with his funding. Marialaura Bancheri prepared the manuscript with contributions from all co-authors. Lastly, the authors declare that they have no conflict of interest.

**Acknowledgments**

The authors acknowledge Trento University project CLIMAWARE
(http://abouthydrology.blogspot.it/search/ label/CLIMAWARE) and European Union FP7 Collaborative Project GLOBAQUA (Managing the effects of multiple stressors on aquatic ecosystems under water scarcity, grant no. 603629-ENV- 2013.6.2.1) that partially financed this research.

## Appendix A: Kriging theory

Kriging is a group of geostatistical techniques used to interpolate the value of random fields based on spatial autocorrelation of measured data, (Isaaks and Srivastava, 1989; Goovaerts, 1997; Kitanidis, 1997). The measurements value $z(\boldsymbol{x}_\alpha)$ and the unknown value $z(\boldsymbol{x})$, where $\boldsymbol{x}$ is the location given according to a certain cartographic projection, are considered as particular

realizations of the random variables $Z(\boldsymbol{x}_\alpha)$ and $Z(\boldsymbol{x})$ (Isaaks and Srivastava, 1989; Goovaerts, 1997). Let the estimation of the (true) random variable $Z(\boldsymbol{x})$ be $Z^\lambda(\boldsymbol{x})$. It is obtained as a linear combination of the $N$ random variables at surrounding points, denoted as $\boldsymbol{x}_\alpha$ with $\alpha = \{1, N\}$, as in Goovaerts (1999):

$$Z^\lambda(\boldsymbol{x}) - m(\boldsymbol{x}) = \sum_{\alpha=1}^{N} \lambda_\alpha(\boldsymbol{x}_\alpha)[Z(\boldsymbol{x}_\alpha) - m(\boldsymbol{x}_\alpha)] \tag{A1}$$

where $m(\boldsymbol{x})$ and $m(\boldsymbol{x}_\alpha)$ are the expected values of the random variables $Z(\boldsymbol{x})$ and $Z(\boldsymbol{x}_\alpha)$; $\lambda(\boldsymbol{x}_\alpha)$ at varying $\alpha$ is the N-uple

of weights assigned to the random variable $Z(\boldsymbol{x}_\alpha)$ at measured sites. The superscript $\lambda$ in $Z^\lambda(\boldsymbol{x})$ denotes that this new random variable is parameterized by the weights. These are chosen to satisfy the condition of minimizing the error of variance of the estimator $\sigma_\lambda^2$, that is:

$$\underset{\lambda}{\mathrm{argmin}}\, \sigma_\lambda^2 \equiv \underset{\lambda}{\mathrm{argmin}}\, Var[Z^\lambda(\boldsymbol{x}) - Z(\boldsymbol{x})] \tag{A2}$$

under the constraint that the estimate is unbiased, i.e.,

$E[Z^\lambda(\boldsymbol{x}) - Z(\boldsymbol{x})] = 0 \tag{A3}$

The latter condition, implies that:

$$\sum_{\alpha=1}^{N} \lambda_\alpha(\boldsymbol{x}_\alpha) = 1 \tag{A4}$$

As shown in various textbooks, e.g., Kitanidis (1997), the above conditions bring to a linear system with the unknown being the N-uple of weights, and the system matrix dependent on the semivariograms (defined below in a simplified case).In synthetic

notation, the linear system can be written as:

$$\Gamma \Lambda = B \tag{A5}$$

where $\Gamma$ is the matrix of two point variograms (defined below), $\Lambda$ is the N-uple of unknown weights and $B$ (the so called known term) is an N-uple containing the variograms between the ungauged site and the measured sites. Further information is required for (A5) to be a solvable linear system. In fact, $B$ is still unknown at this stage.

If isotropy of the spatial statistics of the quantity analyzed is assumed, then the semivariogram is given by, (e.g., Cressie and Cassie (1993)):

$$\gamma(h) = \frac{1}{2N_h} \sum_{i=1}^{N_h} [Z(\boldsymbol{x}) - Z(\boldsymbol{x}_i)]^2 \tag{A6}$$

where $N_h$ denotes the number of observation points at location $\boldsymbol{x}_i$ at distance $h$ from $\boldsymbol{x}$ for any $h$. When random variables are substituted by their available realizations (i.e. $z(\boldsymbol{x}_i)$, indicated with normal letters) an empirical semivariogram is obtained. In order to be extended to any distance, $\gamma$ [h] needs to be fitted to a theoretical semivariogram model, i.e. an assumed function form, as those detailed in Appendix D. The fitting to the theoretical semivariogram model is also necessary to get $B$. In fact, when a theoretical semivariogram is selected, only position information for the ungauged location is required to get its semivariogram with respect to any of the measured locations.

Once B has been determined, the system (A5) can be solved. This procedure is clearly delineated in literature and explained for instance in Kitanidis (1997). Optimized semivariogram models are used to estimate the weighted parameters of kriging algorithm.

## Appendix B:  List of acronyms

| Acronyms | Meaning |
|----------|---------|
| m a.s.l. | meter above sea level |
| CET | central European time |
| DEM | digital elevation model |
| DK | detrended kriging |
| DOI | digital object identifier |
| DP | design patterns |
| GIS | geographical information systems |
| LDK | local detrended kriging |
| LOK | local ordinary kriging |
| LOO | leave-one-out |
| MS | modeling solutions |
| NSE | Nash-Sutcliffe efficiency |
| OMS3 | object modeling system v.3 |
| OK | ordinary kriging |
| PBIAS | percent bias |
| PSO | particle swarm optimization |
| $R^2$ | coefficient of determination |
| RMSE | root mean square error |
| SIK | spatial interpolation kriging |

## Appendix C:  Goodness of fit indices

– Coefficient of determination

The coefficient of determination, $R^2$, is the proportion of variance in the dependent variable that is predictable from the independent variable(s):

$$R^2 = 1 - \frac{\sum_{i=1}^{n}(M_i - S_i)^2}{\sum_{i=1}^{n}(M_i - \overline{M_i})^2} \tag{C1}$$

where $M_i$ is the true value, $\overline{M_i}$ is the mean of $M_i$ and $S_i$ is the predicted value. It varies between 0 to 1, where 1 corresponds to the maximum agreement between predicted and true values.

– Nash - Sutcliffe efficiency

The Nash-Sutcliffe Efficiency (NSE) is a normalized model efficiency coefficient. It determines the relative magnitude of the residual variance compared to the measured data variance (Nash and Sutcliffe, 1970)

$$NSE = 1 - \frac{\sum_{i=1}^{n}(S_i - M_i)^2}{\sum_{i=1}^{n}(M_i - \overline{M_i})^2} \tag{C2}$$

where $S_i$ is the predicted value and $M_i$ is the observed value at a given time step. It varies from $-\infty$ to 1, where 1 corresponds to the maximum agreement between predicted and observed values.

– Percentage bias

Percent bias (PBIAS) measures the average tendency of the simulated values to be larger or smaller than the corresponding measured ones. The optimal value of PBIAS is 0, with small values indicating accurate model simulation. Positive values indicate overestimation bias, while negative values indicate model underestimation bias.

$$PBIAS = 100 \cdot \frac{\sum_{i=1}^{N}(S_i - M_i)}{\sum_{i=1}^{N}M_i} \tag{C3}$$

where $S_i$ is the predicted value and $M_i$ is the observed value.

– Root mean square error

The Root Mean Square Error (RMSE) is given by:

$$RMSE = \sqrt{\frac{1}{N}\sum_{i=1}^{N}(M_i - S_i)^2} \tag{C4}$$

where M and S represent the measured and simulated time-series respectively and N is the number of components in the series.

## Appendix D:  List of semivariogram models implemented in SIK

Using $n$ to represent nugget, $h$ to represent lag distance, $r$ to represent range, and $s$ to represent sill, the 11 theoretical semivariogram models most frequently used in literature are:

– Bessel semivariogram

$$\gamma(h) = s\left(1 - \frac{h}{r}k1\left(\frac{h}{r}\right)\right) \tag{D1}$$

– Circular semivariogram

$$\begin{cases} \gamma(h) = n + s\left\{\frac{2}{\pi}\left[\frac{h}{r}\sqrt{1 - \left(\frac{h}{r}\right)^2}\right] + \arcsin\left(\frac{h}{r}\right)\right\} & h < r \\ \gamma(h) = n + s & h \geq r \end{cases} \tag{D2}$$

– Exponential semivariogram

$$\gamma(h) = n + s(1 - e^{-\frac{h}{r}}) \tag{D3}$$

– Gaussian semivariogram

$$\gamma(h) = n + s[1 - e^{-(\frac{h}{r})^2}] \tag{D4}$$

– Hole semivariogram

$$\gamma(h) = n + s\left[1 - \frac{\sin(\frac{h}{r})}{\frac{h}{r}}\right] \tag{D5}$$

– Linear semivariogram

$$\begin{cases} \gamma(h) = n + s\frac{h}{r} & h < r \\ \gamma(h) = n + s & h \geq r \end{cases} \tag{D6}$$

– Logarithmic semivariogram

$$\gamma(h) = n + s\log\left(\frac{h}{r}\right) \tag{D7}$$

– Pentaspherical semivariogram

$$\begin{cases} \gamma(h) = n + s\left\{ \frac{15}{8}\frac{h}{r} + \left(\frac{h}{r}\right)^3\left[-\frac{5}{4} + \frac{3}{8}\left(\frac{h}{r}\right)^5\right]\right\} & h < r \\ \gamma(h) = n + s & h \geq r \end{cases} \tag{D8}$$

– Periodic semivariogram

$$\gamma(h) = n + s\left[1 - \cos\left(2\pi\frac{h}{r}\right)\right] \tag{D9}$$

– Power semivariogram

$$\gamma(h) = n + sh^r \tag{D10}$$

– Spherical semivariogram

$$\begin{cases} \gamma(h) = n + s\left[1.5\frac{h}{r} - 0.5\left(\frac{h}{r}\right)^3\right] & h < r \\ \gamma(h) = n + s & h \geq r \end{cases} \tag{D11}$$

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
