# Peer review of "The design, deployment and testing of Kriging models in GEOframe with SIK-0.9.8"

_Geoscientific Model Development, 2017_

## Referee Comment (RC1) · Anonymous Referee #1 · 14 Mar 2018

**General Comments**

*Overall quality of the paper*

Firstly, thanks the authors for this manuscript that delivering a new software product to science world is important for diversity and productiveness. In addition, I want to remark that the review is to improve the paper.

The paper presents an interpolation package for climatic and environmental variables. The objectives of the paper are to bring an alternative package which is applicable with good performance. By using Kriging techniques, the modeling spatial distribution of hourly temperature and rainfall in an Alpine catchment located in north of Italy and leave-one-out cross validation of the model are the scope of study. And it is suitable in terms of themes of GMD journal.

The paper is generally clear and well-written; the methods, assumptions and results are rationally structured and presented, but I think the language has some issues in the sense of fluency. In the manuscript, the structure of parenthetical sentence is overutilized. Instead of this structure, simple and short sentences with proper conjunctions must be preferred for fluency and explicity of text. Clarity of the manuscript can be enhanced a bit more by using clear, fluent and open expressions. In addition, I do not approve the separation of word with a hyphen at the end of the line.

In my opinion, the paper is at good level in terms of significance, quality and reproducibility of scientific and presentation quality has a fair level. I think it could be accepted after revision of the manuscript.

Advantages of the manuscript:

- A new open source software product
- Comparison of 4 types of Kriging and 11 semivariogram models
- Comparison with R gstat in terms of multi-model approach

Disadvantages of the manuscript:

- The lack of multi-site (which have different geo-climatic characteristics) comparison
- Independence of validation dataset
- The insufficiency of rainfall data used (the modeling of a single storm event does not provide insight into the good performance)

**Specific Comments**

*Major-minor issues (criticisms), review based sections, clarification of figures and tables, evaluation of references, questions*

Against the possibility of future modifications, the title must include version number (v. 0.9.8) of the model.

In the abstract, time scale (hourly) of used data could be mentioned. Also, "2008 year hourly temperature", "rainfall storm event (11 h) in 2008" and "spatio-temporal distribution of climatic and environmental variables" expressions could be included in the abstract. The importance of the study for hydro-meteorological studies can be processed in some more detail in the abstract and the introduction.

A bit more of recent references on Kriging as itemized below can be given in introduction section. Moreover, references on Kriging open source algorithms can be added and it is provided the differences of between this paper and the current literature.

- Saghafian and Bondarabadi, 2008. "Validity of regional rainfall spatial distribution methods in mountainous areas".
- Di Piazza et al., 2011. "Comparative analysis of different techniques for spatial interpolation of rainfall data to create a serially complete monthly time series of precipitation for Sicily, Italy".
- Bostan et al., 2012. "Comparison of regression and kriging techniques for mapping the average annual precipitation of Turkey".
- Wang et al., 2014. "Comparison of interpolation methods for estimating spatial distribution of precipitation in Ontario, Canada".
- Adhikary et al., 2016. "Genetic programming-based ordinary kriging for spatial interpolation of rainfall".
- Jin et al., 2016. "Estimating Loess Plateau average annual precipitation with multiple lineer regression kriging and geographically weighted regression kriging".

As a reference, the study of titled "A statistical approach to some mine valuations and allied problems at the Witwatersrand" by Krige (1951) which is the originator of Kriging method, can be quoted within the framework of respect for labor. As you know, Kriging method is named after him.

Kriging methods are geostatistical methods that consider the spatial autocorrelation of sample data. Due to based on stochastic process, Kriging assumes that sample data is stationary and normally distributed. And the data needs the de-trending and de-clustering processes. In this context, fit of normal distribution and autocorrelation function are analyzed. Normal distribution and spatial autocorrelation of residuals are even investigated. Within this context, detailed information about the processes of outlier treatment, normalization, detrending, declustering, optimization of semivariogram model and neighborhood search must be provided.

In introduction, some open source geostatistical tools are given. In addition to them, software products such as;

- GRASS (grass.osgeo.org)
- GSLIB (www.gslib.com)
- Surfpack (dakota.sandia.gov/content/surfpack)
- jk3d (sourceforge.net/projects/jk3d)
- Map Window (mapwindow.org)
- SGeMS (sgems.sourceforge.net)
- GitHub Py Krige (github.com/bsmurphy/PyKrige)
- QGIS (qgis.org)
- uDig (udig.refractions.net)
- GeoDa (spatial.uchicago.edu/software)
- ArcGIS (arcgis.com) -even it can be counted from the point of view of integrating open source codescan be indicated.

The study contains continuous modeling of temperature and event based modeling of rainfall by SIK package. Temperature outputs are also compared with R gstat. Although temperature is more stabile climatic variable in according to the spatial distribution, precipitation is more changeable that has more the spatial variance. Thus, I think continuous modeling and comparison with different package results of precipitation make greater contribution to this paper.

It is stated that plug-in to a hydrological model of the SIK model is easy. Entering more details about this will be good. Furthermore, the SIK results can be inputted into a hydrological model and the performance of hydrological model can be assessed.

In some parts of the paper, it is specified that the data of 97 meteorological stations is used, but there are 93 stations in Appendix D. More than this, there are 81 stations in Figure 4; 57 stations in Figure 8. In short, an inconsistency in number of stations is in existence. When examining Figure 8, I have doubts about a manipulation for high performance. Please enlighten me about this matter.

To get to the clarification of figures, I think Figure 3 and 5 must be made smaller, due to they are nonsensically very big. In Figure 4, basin boundary and river network should be shown for more understanding of the study area. In the legend of Fig. 4, the ranges of elevation must be like "210.1-850, 850-1500, 1500-2100, 2100-2750, 2750-3388" with round numbers as an example. For Figure 5 and 9, please give an explanation of X and Y axis in the relevant places of the text. In the legend of Figure 5, please illustrate experimental semivariance as only black dot without line and the others as only line without dot. In Figure 6, size of dots can be smaller in terms of clearly view of each semivariogram model. A scale bar (km) must be added on Figure 7 where the intervals of temperature must be as "-6-0, 0-5, 5-10, 10-15, 15-19" with round values. Likewise, A scale bar and north arrow must be placed in Figure 8 which has two same legends of elevation, one of them must be removed. In comparison of spatial data, same color scale must be used for spatial layers. In this respect, Figure 7 (temperature scale) and 8 (elevation scale) are good examples. However, sizes of RMSE plots

are not same in Figure 8. In despite of the authors allege the visualization reasons as excuse, I do not accept this situation. For a reasonable comparison, same size scale of RMSE must be used. I suggest the followed intervals for RMSE in both OK and DK output layers of June 2008. "0-1, 1-2, 2-3, 3-6, 6-9, >9 with max value". A conspicuous another issue about Figure 8 is that max RMSE value is 9 in spite of given as 11.95 in the text. The measured rainfall is not shown as black dashed line in the legend of Figure 10. In Figure 11, the color scale of precipitation is changed from white to dark blue. But the colors of precipitation classes are very close to each other, because of this situation the transition of precipitation surface is not distinguished exactly. To remove this problem, selection of a more colorful scale will be suitable. For instance; red, yellow, sea green, light and dark blue, purple. It changes respectively from dry and wet. Two raster layers (DEM and precipitation), both of which contain white color, are overlapped using transparency feature of the layers in the same figure; but this is caused the confusion with regard to understanding of the figure. If the presentation of elevation layer is as contour lines, the confusion dies. The intervals of the precipitation and DEM must be respectively as "2.21-2.7, 2.7-3.2, 3.2, 3.7, 3.7-4.2, 4.2-4.7, 4.7-5.2, 5.2-5.7, 5.7-6.02" and "0-300, 300-2900, 2900-3388". Figure 11 also needs a scale bar. Together with NSE; $R^2$, RMSE and PBIAS values of DK and similarly all the performance criteria of OK methods can be given in Figure 12. And all, the captions of the figures can be shortened as follows and the required explanations relating to the figures should be stated in the relevant places of the text. In all the figures, writings and titles of chart and axis must be had standard font size.

- Figure 1. Flow chart of modeling process.
- Figure 2. Flow chart of validation process.
- Figure 1. Flow chart of modeling and validation processes. (Figure 1 and 2 can be combined into one)
- Figure 4. Geo-location of study area and position of meteorological stations.
- Figure 5. Fitting of the experimental semivariogram for 15th June 2008 12:00 CET.
- Figure 6. Monthly variation of the NSE index over entire hourly temperature dataset.
- Figure 7. Maps and histograms of spatialized temperature for 15th February 2008 and 15th June 2008.
- Figure 8. Bubble plots of the RMSE obtained using OK and DK.
- Figure 9. Boxplots of the semivariograms of the precipitation event of 29th and 30th June 2008.
- Figure 10. Comparison between the four types of Kriging and the measured rainfall.
- Figure 11. Spatial interpolation of the precipitation applying OK and the linear semivariogram model.

A little more information such as slope, land cover, vegetation, hypsometric elevation and river discharge about the study area can be given. Basin characteristics are important in understanding and evaluation of hydro-meteorological modeling in particular type and distribution of precipitation for this study. Info of DEM is not presented in the paper. Please provide source of DEM.

Please give definitions, explanations and formulations of the performance criteria used (NSE, $R^2$, RMSE, PBIAS) in the paper. For NSE, it is referred to Nash and Sutcliffe (1970).

The headers of the tables can be changed as below because of they are also overlong. The statements about the tables can be presented in the relevant parts of the text not in the table header. It is emphasized that Bessel semivariogram model is the best in according to Table 1. How was this information reached? How was the integration of 4 criteria (NSE, $R^2$, RMSE, PBIAS) achieved? Logarithmic model results also look pretty good, why was not the logarithmic model in the best 5 of the authors? How were the best 5 determined? In my opinion, the logarithmic model is candidate for the best and Bessel, logarithmic, exponential, linear and spherical models are the best 5 semivariograms. Please clarify this matter. In Table 2, mean of performance criteria values of all the stations can be given for 4 types of Kriging.

- Table 1. Performance results of semivariogram models used
- Table 2. Results in terms of goodness of fit indices between the measured and interpolated rainfall values for two stations

In the manuscript, it is stated that the overall performances of both SIK and gstat tools are good. I don't think gstat has good performance in according to Figure 12. Please research the success limits of NSE values in the literature. In this context, please rewrite the related parts and the last sentence of the abstract. In plain words, I did not expect it to be so different. SIK has almost overwhelming superiority. I have some suspicions in this regard. Please inform about the optimization of gstat, is it performed sensitively? How do you describe this failure and explain it?

In conclusion section, resolution of interpolation grid data (100 m) can be mentioned because of the resolution is an important issue for the hydrological model results.

The following abbreviations and symbols must be added to Appendix B:

- PSO: particle swarm optimization
- OK: ordinary kriging
- LOK: local ordinary kriging
- DK: detrended kriging
- LDK: local detrended kriging
- SIK: spatial interpolation kriging
- OMS3: object modeling system v.3
- MS: modeling solutions
- DP: design patterns
- LOO: leave-one-out
- GIS: geographical information systems
- a.s.l. : above sea level
- CET: central European time
- DEM: digital elevation model
- DOI: digital object identifier

- $\Gamma$: matrix of the two point variograms
- $\Lambda$: N-uple of the unknown weights
- B: N-uple containing the variograms among the ungauged site and the measured sites

In most of the equations in Appendix C, multiplication is indicated by a dot. I think there is no need to use these dots.

In Appendix D, sorting of the stations by elevation not station ID becomes more meaningful due to the strong relationship between the elevation and temperature or precipitation. Giving of mean and median elevation values of the all stations would be better.

In appendix of code availability, the authors specifies that the documentation of SIK components are available. Is a detailed user manual provided?

The sentence of "Lastly, the authors declare that they have no conflict of interest." should be placed at the end of author contribution appendix.

**Technical Corrections**

*Corrections, typing errors*

In some parts of the paper, it is specified that the data of 97 meteorological stations and 10 semivariograms are used, but there are 93 stations in Appendix D and 11 semivariogram models in Figure 5/Table 1.

**P1 L7-P8 L14- P11 L8- P20 L18:** "97" ---> "93"

**P1 L5-P20 L20-P22 L2:** "ten" ---> "eleven"

**P10 L6-P10 L14-P11 in Fig. 5 caption-P11 L2-P12 in Table 1 header-P20 L2:** "10" ---> "11"

**P1 L3:** "in a hydrological model" ---> "to a hydrological model"

**P1 L8:** "compared to the results" ---> "compared to the results of temperature"

**P1 L11+:** "These data, besides being important by themselves, are the natural input to distributed and semi-distributed hydrological models, where their quality and precision affect the accuracy of results." ---> Please give a few references.

**P1 L15+-P1 L19-P1 L22-P3 L5+-P5 L5-P7 L17:** The references aren't chronologically presented, please chronologize.

**P1 L18:** "anong" ---> "among"

**P1 L20:** "shown" ---> "showed"

**P2 L25:** "(Formetta et al., 2014).In order" ---> "(Formetta et al., 2014). In order". No space after point.

**P3 L13-P3 L15-P3 L27:** The ordering of in-equation parenthesis is not appropriate. The correct ordering is "{ [ ( ) ] }".

**P3 L9-P3 L22-P 3 L23:** "N-ple" ---> "N-uple"

**P3 L26-P5 L21-P7 L11:** "e.g." ---> "e.g.,"

**P3 L27:** ":=" ---> "="

**P4 L2-P4 L3:** "i.e." ---> "i.e.,"

**P4 L19:** "are the local" ---> "are respectively the local"

**P4 L28:** "interpolate a theoretical model for the semivariogram" ---> "fit a theoretical model to the semivariogram"

**P4 L30:** "produce maps" ---> "produce continuous surface maps"

**P5 L7:** "Efron and Efron" ---> "Efron"

**P5 L10:** "produced" ---> "performed"

**P5 L16:** "granularity whose rational" ---> "detail whose logic"

**P5 L17:** "that, for the reader convenience are presented in A" ---> "presented in Appendix A for the reader convenience"

**P5 L19:** "Object Modelling System v3 (OMS3)" ---> "OMS3". Previously presented as open.

**P5 L20:** "here" ---> "are"

**P5 L22:** "Modelling Solutions (MS)" ---> "MS". Previously presented as open.

**P5 L22:** "complicate" ---> "complicated"

**P5 L25:** "particular cares from the researcher who programs them" ---> "any special effort of the computer programmer of them"

**P5 L29+:** "in various components, and inserted in MS as alternatives, thus opening the way to compare, inside the same chain of tools" ---> "various components inserted in MS as alternatives. Thus inside the same chain of tools, it enables to compare"

**P5 L31:** "in Formetta et al. (2014) and in Bancheri (2017)" ---> "Formetta et al. (2014) and Bancheri (2017)"

**P6 L12:** "Particle swarm" ---> "Particle swarm optimization (PSO) technique"

**P6 L16-P11 L7-P14 L2-P17 L2:** "show" ---> "shows"

**P7 L10:** "addition(" ---> " addition (". No space before parenthesis.

**P7 L11:** "Design Patterns (DPs," ---> "DP". Previously presented as open.

**P7 L13-P7 L14-P7 L15:** "DPs" ---> "DP"

**P8 L13-P20 L11:** "modeling solutions" ---> "MS". Previously presented as open.

**P9 in Fig. 4-P18 in Fig. 11:** "Meteo_stations" ---> "Meteorological stations"

**P9 in Fig. 4:** "DEM" ---> "DEM (m)"

**P18 in Fig. 11:** "DEM" ---> "Contour lines (m)". In Fig. 11, contour lines must be used instead of DEM.

**P9 L9:** "out-layers" ---> "outliers"

**P9 L10-P10 L3:** "asses" ---> "assess"

**P10 L1+:** "nor too low or too high" ---> "neither too low nor too high"

**P10 L10:** "their performances" ---> "performances"

**P11 L1:** "the correlation coefficient" ---> "the coefficient of determination"

**P11 L5:** "4 typer of Kriging, " ---> "4 types of Kriging"

**P11 L9:** It is specified that the two local cases were performed using the ten closest stations to the interpolation point. Why 10 stations? Has neighborhood analysis ever carried out?

**P11 L10:** "meaning" ---> "it means"

**P11 L12:** "exists" ---> "exist"

**P12 in Table 1:** Please sort the semivariograms by the performance.

**P13 in Fig. 6:** Please remove expression of "hourly temperature" from the chart titles as it is expressed in the caption.

**P14 L2:** "7" ---> "Figure 7"

**P14 L3:** "during" ---> "in"

**P15 L1:** "alms" ---> "m a.s.l."

**P15 L2:** "1.83 $^{o}$C" ---> "1.83 $^{o}$C in the DK method"

**P15 in Fig. 8:** Please remove expression of "monthly mean temperature" from the map titles. The altitude legend scale has not min and max values, please add these.

**P16 in Fig. 9 chart title:** "30 h" ---> "11 h"

**P16 L2:** "953 m" ---> "943 m". According to Appendix D.

**P16 L6:** "local ordinary kriging" ---> "LOK". Previously presented as open.

**P16 L7:** "ID 2170" ---> "ID 1270"

**P16 L6-P16 L8:** It is stated that the LOK computed using the 5 closer stations. Is it determined with manual calibration of the neighborhood? If search of neighbor numbers was performed, it must be specified in the text. If not, explain that how it was determined. In addition to, the number of closest neighbor stations is 10 for temperature and 5 for rainfall. Please interpret this.

**P17 L1:** "DEM (100 m resolution)" ---> "DEM"

**P17 L3:** "[mm/h]" ---> "(mm/h)"

**P19 L4:** "design pattern" ---> "DS". Previously presented as open.

**P19 L4:** "figure 3" ---> "Figure 3"

**P19 L7:** "opertion" ---> "operation"

**P19 L10:** "offers" ---> "offer"

**P19 L11+:** "the same number of bins and the same cutoff distance" ---> "the same number of bins and cutoff distance"

**P19 in Fig. 12 legend:** "SI component" ---> "SIK component"

**P19 in Fig. 12 caption:** "package" ---> "packages"

**P19 in Fig. 12:** NSE value 1.0 is not found in the figure.

**P20 L13:** "design patterns" ---> "DS". Previously presented as open.

**P20 L18+:** "Both the interpolation of the temperature and the interpolation of the rainfall" ---> "Both the interpolation of the temperature and the rainfall"

**P20 L22:** "detrended Kriging" ---> "DK". Previously presented as open.

**P20 L22:** "Single event rainfall, on the contrary do not" ---> "On the contrary, single event rainfall did not"

**P20 L26:** "GPL v3" ---> "GPL v3 (www.gnu.org/licenses/gpl-3.0.en.html)"

**P20 L27:** "a a-personal" ---> "a-personal"

**P20 L28:** ""

**P21 L1:** "instead, for instance, of Mercurial," ---> "instead of for instance Mercurial"

**P21 L1:** "found at" ---> "found at followed link"

**P21 L5:** "includes" ---> "include"

**P21 L7:** "Maven and Gradle, in particular" ---> "in particular Maven and Gradle"

**P21 L7+:** "Our favor, among the possibilities, was for" ---> "Our favor among the possibilities was"

**P21 L12:** "instead than another All of them" ---> "than another. All of them"

**P21 L13:** "programmers' community, but are" ---> "community of programmers, but they are"

**P21 L13+:** "scientists making increasingly difficult for them maintain their own code" ---> "scientists who are increasingly struggling against difficulties to maintain their own code"

**P21 L18:** "and run all the software" ---> "and all the software"

**P21 L20:** "GPL v3 license (www.gnu.org/licenses/gpl-3.0.en.html)" ---> "GPL v3 license"

**P21 L22:** "alternatives are, among others," ---> "alternatives among others are"

**P21 L25:** "our" ---> "ours"

**P22 in Appendix B:** "Percent bias" ---> "percent bias"

**P22 in Appendix B:** "Coefficient of determination" ---> "coefficient of determination"

**P22 in Appendix B:** "Root mean square error" ---> "root mean square error"

**P25 in Appendix D:** The elevation of station 90145 is 3399 m as given in Appendix D and P15 L1. However the max elevation value of study area is 3388 m in Fig. 4. Here, there is a mistake (3399>3388), please correct it.

**P29 L5:** "Fisheries research" ---> "Fisheries Research"

**P29 L25-P29 L37-P30 L10:** "Oxford university press" ---> "Oxford University Press"

**P29 L29:** "Efron, B. "

**P29 L36: "**Springer Publishing" must be added.

**P30 L2:** "Journal of hydrology" ---> "Journal of Hydrology"

**P30 L7:** "Journal of applied meteorology" ---> "Journal of Applied Meteorology"

**P30 L8:** "International journal of geographical information systems" ---> "International Journal of Geographical Information Systems"

**P30 L21:** "Biotechnol. Agron. Soc. Environ., vol. 17(2), 392-406" must be added.

**P30 L27:** "Mathematical geology" ---> "Mathematical Geology"

**P30 L31:** "Publ. Climatol., 46" must be added.

**P30 L35:** "JAWRA Journal of the American Water Resources Association, 21(3), 365–380" must be added.

**P30 L36:** "Monthly weather review" ---> "Monthly Weather Review"

The references in **P30 L22** and **P30 L24** must be replaced due to the chronological order.

**P31 L3:** "Agricultural systems" ---> "Agricultural Systems"

**P31 L6:** "Data Acquisition And Processing, Analysis, Forecasting And Other Applications, (WMO No.168)" must be added.

**Recommendations**

- The validation of SIK component can be performed with remote sensed data (radar and satellite products) besides of ground based data (earth observation stations).
- The climatic and environmental variables can be analyzed at different time scales as daily, monthly, seasonal and annual.
- The interpolation performance can be evaluated in different areas in point of multi-site approach.
- Co-Kriging using secondary data as elevation, vegetation and temperature can be embodied into the SIK.
- Kling-Gupta Efficiency (KGE) criteria whose advantages over NSE are stated in Gupta et al., 2009, can be added as performance benchmark.
- Interpolated temperature and rainfall data can be inputted into a hydrological model. And accuracy and reliability of interpolation can be tested in terms of performance of hydrological model.
- In assessment of interpolation performance, jack knife and k-fold cross validation can be utilized for independent data set of validation procedure.
- Fit of normal distribution of residuals can be tested due to Kriging modeling is a stochastic process.
- Uncertainty analysis can be carried out.
- Comparison based on pixels of raster layer can be implemented.
- Comparison in terms of lowland-highland, underestimate-overestimate, density of stations and representation of low-high rainfall.

**References**

Adhikary, S. K., Muttil, N. and Yilmaz, A. G., 2016. Genetic programming-based ordinary kriging for spatial interpolation of rainfall. *J. Hydrol. Eng.*, 21 (2), -1--1.

Bostan, P. A., Heuvelink, G. B. M. and Akyurek, S. Z., 2012. Comparison of regression and kriging techniques for mapping the average annual precipitation of Turkey. *Int. J. of Applied Earth Observation and Geoinformation*, 19, 115-126.

Di Piazza, A., Lo Conti, F., Noto, L. V., Viola, F. and La Loggia, G., 2011. Comparative analysis of different techniques for spatial interpolation of rainfall data to create a serially complete monthly time series of precipitation for Sicily, Italy. *Int. J. of Applied Earth Observation and Geoinformation*, 13, 396-408.

Gupta, H. V., Kling, H., Yilmaz, K. K. and Martinez, G. F., 2009. Decomposition of the mean squared error and NSE performance criteria: Implications for improving hydrological modelling. *Journal of Hydrology*, 377, 80-91.

Jin Q., Zhang J., Shi M. and Huang J., 2016, Estimating Loess Plateau average annual precipitation with multiple lineer regression kriging and geographically weighted regression kriging. *Water*, 8, 266, 1-20.

Krige, D. G., 1951. A statistical approach to some mine valuations and allied problems at the Witwatersrand. *MSc Thesis of the University of Witwatersrand.*

Nash, J. E., and Sutcliffe, J. V., 1970. River flow forecasting through conceptual models. Part I-A discussion of principles. *Journal of Hydrology*, 10(3), 282-290.

Saghafian, B. and Bondarabadi, S. R., 2008. Validity of regional rainfall spatial distribution methods in mountainous areas. *J. of Hydrol. Eng.*, 13 (7), 531-540.

Wang, S., Huang, G. H., Lin, Q. G., Li, Z., Zhang, H. and Fan, Y. R., 2014. Comparison of interpolation methods for estimating spatial distribution of precipitation in Ontario, Canada. *Int. J. Climatol.*, 34: 3745-3751.

---

## Referee Comment (RC2) · Anonymous Referee #2 · 15 Mar 2018

Referee Report The design, deployment and testing of Kriging models in GEOframe Bancheri M., et al. The technical paper presents a package for the interpolation of climatological variables using kriging techniques. The structure of the package is described, and the results of the application to two datasets (a rainfall and a temperature one) are presented. The authors also provide a comparison with the R package gstat. The technical paper aims at describing a package that could be of interest for the geoscientific community due to the fact that it is specifically designed for the interpolation of climatological variables and is suitable to be easily merged into hydrological models. The manuscript then would fit the purposes of the journal, if focused on the description of this package. Despite this, the aim of the paper is not clear. On the one hand, reading the abstract, the expected focus is the tool itself: its capability and the novelty it

introduces in comparison with the available software. On the other hand, going through the paper one can realize that just a limited part of the work is focused on the package, and the core of it is focused on the analysis of the results obtained with the different tested kriging methods. Considering that a wide geostatistical literature focused on the assessment of the quality of the interpolation of climatological variables already exists (e.g., Hartkamp et al., 1999, Moral F. J., 2010, etc.), and that the analysis of the performance of the package are limited to sect.6, I think that the authors should adjust the focus of the whole manuscript (or at least clarify their aims) before it can be accepted for publication. A series of major comments follows, all related to this topic, and some minors ones are added, that I think the authors should address before the manuscript is ready for publication. Major comments Most of the manuscript is dedicated to the kriging methodology. The use of kriging methodologies is quite established in the hydrological literature and many studies on the performance of the different kriging methods can be found. The core part of the paper should be the description of the tool and the assessment of its potentialities. In the detail: 1. My opinion is that the authors dedicate too much space at the description of the theoretical basis of the kriging (Sect. 2) while some references and a general outline would be enough. 2. The analysis of the results of the case study turns out to be the core of the paper (Sect. 5 is the longest one with 6 out of 12 figures dedicated). Section 5 should be revised, shortened, and the number of related figures should be reduced. The interest of the case study is in checking the functionality of the package rather than to carry out an analysis of the performances of the various interpolators 3. While the structure of the tool is clearly explained in sect. 3 the analysis of the performance, compared to other software, is not sufficient. As the authors themselves state, it is an important point "to justify the introduction of an alternative software". The authors just refer to general "way longer" (L.5 P.19) computational time of gstat compared to SIK, and a comparison of the NSE for a single model (figure 12). This does not appears enough for a scientific paper: the authors should provide numbers and figures to support and highlight performances of their package in comparison with other software. Their opinion of "greater ease" (L.8 P.19) is not enough,

considering that gstat turns out to be more versatile under some points of view (e.g., it considers both omnidirectional and directional variograms) and that there are some other kriging packages offering a user-friendly and easy-to-manage interface (e.g., the v.kriging command in GRASS-Gis). Minor comments 4. Abstract: the abstract should be reviewed according to the modification of the structure of the paper. At the moment it does not describe the focus of the work. 5. L.9, P.2 – It must be pointed out that GRASS-GIS is also available as an open-source GIS offering a user-friendly (v.kriging) function for performing kriging 6. Variables should be named always in italic (e.g., "B" L.22, P.3), and measurement units not in italic (e.g., km2 L.1, P.9) 7. Figure 1 and 2 can be merged together as they differ only for the last block. 8. APPENDICES: I think the authors make a misuse of the appendices. Appendix A could be merged in the main part of the paper as deal with an important phase of the development of the package and Appendix D is almost useless for the aims of the paper (the table should be provided as supplementary material). Reducing the number of the symbols also Appendix B would be useless, and the meaning of the symbol could be inserted along the main text. 9. Writing style and use of English could be improved: the manuscript would benefit from extensive English editing by a native speaker. The text should also be double-checked for typos and other minor errors (a non-exhaustive list follows): o L.18, P.1. "anong" -> "among" o L. 8, P.2. There is a useless open bracket o L. 4, P. 4. "Appendix A" -> "Appendix C" o L. 17, P. 5 I don't think "contemporary" is a proper word in this case o L. 10, P. 8 "classes" -> "class" o L. 14, P. 8 "...in D" -> "..in Appendix D" o L. 7, P. 10 "where" -> "were" 10. Reference style is not uniform, sometimes brackets include only the year (e.g., L.1, P.2) and sometimes name and year (e.g., L.25, P.2). Please adopt the journal style. References Hartkamp, A. D., De Beurs, K., Stein, A., & White, J. W. (1999). Interpolation techniques for climate variables. Moral, F. J. (2010). Comparison of different geostatistical approaches to map climate variables: application to precipitation. International Journal of Climatology, 30(4), 620-631.

---

## Author Comment (AC1) · 19 Apr 2018

The authors thank the reviewer for her/his observations, which helped us to improve our manuscript. This response does not yet produce a new manuscript, which will be submitted after the closure of the review phase, upon request of the Editor. We introduce here, however, the main adjustments that we will make in the final version of the revised paper based upon the reviewer's suggestions.

General Comments Overall quality of the paper

R1: Firstly, thanks the authors for this manuscript that delivering a new software product to science world is important for diversity and productiveness. In addition, I want to remark that the review is to improve the paper. The paper presents an interpolation

package for climatic and environmental variables. The objectives of the paper are to bring an alternative package, which is applicable with good performance. By using Kriging techniques, the modeling spatial distribution of hourly temperature and rainfall in an Alpine catchment located in north of Italy and leave-one-out cross validation of the model are the scope of study. And it is suitable in terms of themes of GMD journal.

A1: We thank the reviewer for this recognition.

R2: The paper is generally clear and well-written; the methods, assumptions and results are rationally structured and presented, but I think the language has some issues in the sense of fluency. In the manuscript, the structure of parenthetical sentence is overutilized. Instead of this structure, simple and short sentences with proper conjunctions must be preferred for fluency and explicity of text. Clarity of the manuscript can be enhanced a bit more by using clear, fluent and open expressions. In addition, I do not approve the separation of word with a hyphen at the end of the line.

A2: The manuscript will go through an extended revision of the English grammar and fluency

R3: In my opinion, the paper is at good level in terms of significance, quality and reproducibility of scientific and presentation quality has a fair level. I think it could be accepted after revision of the manuscript.

Advantages of the manuscript: A new open source software product Comparison of 4 types of Kriging and 11 semivariogram models Comparison with R gstat in terms of multi-model approach

Disadvantages of the manuscript: The lack of multi-site (which have different geoclimatic characteristics) comparison Independence of validation dataset The insufficiency of rainfall data used (the modeling of a single storm event does not provide insight into the good performance)

A3: The scope of the paper is to present a new geostatistical tool, which could be a

good competitor of other open-source software. The strength of our product is not only its capability to do a good Kriging interpolation but also the possibility to be used as stand alone or plugged to a hydrological model. We believe also that the engineering of our software (an aspect usually neglected in scientific reports) could be of interest for GMD readers. As stated in some of the following answers, a deeper assessment of its performance has already carried out in other works by the Authors, (Formetta et al. 2014, Abera et al. 2017). The two previous works include multi-site comparisons, modelling of different rainfall dataset and the assessing of the impact of the type of Kriging and theoretical semivariogram model on the mean annual runoff. Therefore performing here these analysis would not add very much to literature.

Specific Comments Major-minor issues (criticisms), review based sections, clarification of figures and tables, evaluation of references, questions.

(1) Against the possibility of future modifications, the title must include version number (v. 0.9.8) of the model.

The number of the version, as the reviewer said, is an important information that should be added. However, since in the title no reference is made to the SIK package, we decided to add it in the Introduction Section.

(2) In the abstract, time scale (hourly) of used data could be mentioned. Also, "2008 year hourly temperature", "rainfall storm event (11 h) in 2008" and "spatio-temporal distribution of climatic and environmental variables" expressions could be included in the abstract.

We added the suggested information about the spatio-temporal distribution of the used data.

(3) The importance of the study for hydro-meteorological studies can be processed in some more detail in the abstract and the introduction.

We added more references and details about the impact of the interpolation on hydro-

meteorological studies in the introduction. In particular we added the following sentences and references, using also other reviewer's comments: " These data, besides being important by themselves, are the natural input to distributed and semi-distributed hydrological models. Their quality and precision affect the accuracy of results, (Xu and Singh, 1998; Stooksbury et al., 1999; Balme et al., 2006; Abera et al., 2017). In fact, all the surface waters models require a reliable precipitation dataset that is complete both in space and in time. Quite often, however, datasets of hydrological variables suffer from errors and missing data, therefore, filling the gaps in time series by estimating the missing values is a common approach to solve this problem, (Eischeid et al., 2000; Saghafian and Bondarabadi, 2008; Di Piazza et al., 2011; Adhikary et al., 2015)."

(4) A bit more of recent references on Kriging as itemized below can be given in introduction section. Moreover, references on Kriging open source algorithms can be added and it is provided the differences of between this paper and the current literature.

Saghafian and Bondarabadi, 2008. "Validity of regional rainfall spatial distribution methods in mountainous areas". Di Piazza et al., 2011. "Comparative analysis of different techniques for spatial interpolation of rainfall data to create a serially complete monthly time series of precipitation for Sicily, Italy". Bostan et al., 2012. "Comparison of regression and kriging techniques for mapping the average annual precipitation of Turkey". Wang et al., 2014. "Comparison of interpolation methods for estimating spatial distribution of precipitation in Ontario, Canada". Adhikary et al., 2016. "Genetic programming-based ordinary kriging for spatial interpolation of rainfall". Jin et al., 2016. "Estimating Loess Plateau average annual precipitation with multiple linear regression kriging and geographically weighted regression kriging".

As a reference, the study of titled "A statistical approach to some mine valuations and allied problems at the Witwatersrand" by Krige (1951) which is the originator of Kriging method, can be quoted within the framework of respect for labor. As you know, Kriging method is named after him.

All the previous references are really interesting and worth to be mentioned in the paper. Some of the references were added in the Introduction section, as shown in the previous answer. However, since the principle aim of our work is not to asses the Kriging interpolation but to present a new package, open source, Java-based, well integrated in a hydrological model, in the perspective of reproducible research, a comparison between the previous papers and our is out of the scope of the paper.

(5) Kriging methods are geostatistical methods that consider the spatial autocorrelation of sample data. Due to based on stochastic process, Kriging assumes that sample data is stationary and normally distributed. And the data needs the de-trending and de-clustering processes. In this context, fit of normal distribution and autocorrelation function are analysed. Normal distribution and spatial autocorrelation of residuals are even investigated. Within this context, detailed information about the processes of outlier treatment, normalization, detrending, declustering, optimization of semivariogram model and neighbourhood search must be provided.

One of the major comments of the Referee #2 is that we dedicated too much space to the description of the theoretical basis of the kriging and to the analysis of the results in Section 5. Therefore, according to his/her reviews, we will try to shorten the theoretical and the result parts, to give more space to the informatics description. Some of the previous information (treatment of the outliers, neighbourhood search, optimization of semivariogram) are already given in the related sections of the paper and, according to us, add further parts would make the paper less readable and our contents less effective. For example, in the Setup subsection the description of the treatment of the outliers, the choice of the number of bins of distances, the information on the neighbourhood search were discussed.

(6) In introduction, some open source geostatistical tools are given. In addition to them, software products such as; GRASS (grass.osgeo.org) GSLIB (www.gslib.com) Surfpack (dakota.sandia.gov/content/surfpack) jk3d (sourceforge.net/projects/jk3d) Map Window (mapwindow.org) SGeMS (sgems.sourceforge.net) GitHub Py Krige

(github.com/bsmurphy/PyKrige) QGIS (qgis.org) uDig (udig.refractions.net) GeoDa (spatial.uchicago.edu/software) ArcGIS (arcgis.com) -even it can be counted from the point of view of integrating open source codes -can be indicated.

The Authors thank the reviewer for this list of alternative software. Some of these packages are actually open source, easy to import into a custom script and actively developed. Between them, it is worth mentioning: âŮŔ PyKrige: a documented and actively developed python package. It covers both 2D and 3D ordinary and universal Kriging; âŮŔ Dakota (Surfpack): well documented and actively developed C++ software. These represent very good competitors of the SIK package, as well as gstat. However, SIK is the only Java and component-based software. QGIS and GRASS (v.kriging) krigings are both encapsulated into GIS software and are easy to include into scripts leveraging GIS capabilities. Making them part of a modeling solution is pretty cumbersome though. Map Windows is yet open source, available on github and actively developed but it is GIS-based and mainly Windows OS oriented. The other packages are old, hardly accessible or poorly documented. Eventually, ArcGIS is not open source.

(7) The study contains continuous modeling of temperature and event based modeling of rainfall by SIK package. Temperature outputs are also compared with R gstat. Although temperature is more stable climatic variable in according to the spatial distribution, precipitation is more changeable that has more the spatial variance. Thus, I think continuous modeling and comparison with different package results of precipitation make greater contribution to this paper.

According to the reviewer suggestions, we added also a comparison of the results of precipitation interpolation.

(8) It is stated that plug-in to a hydrological model of the SIK model is easy. Entering more details about this will be good. Furthermore, the SIK results can be inputted into a hydrological model and the performance of hydrological model can be assessed.

Regarding the easiness of the SIK package to plug-in to a hydrological model, we added more information in the Introduction and in the Design of the SIK package section. In particular, in the Introduction, the following sentences were added: " SIK inherits some previous code used, for instance, in Formetta et al. (2014) and Abera et al. (2017). In particular Abera et al. (2017) assessed the effect of the interpolation of precipitation on long-term mean annual runoff. To make the code more flexible, easily extensible, maintainable, SIK was completely refactored and a systematic use of Design Patterns (DP), (Gamma, 1994; Freeman et al.,2004), was introduced. " As stated in the previous sentences, the assessment of the performances on the SIK package on the hydrological results has already been presented in Abera et al. 2017, where a previous version of the component, based on the same algorithms, was used. The main difference between that version and the present one is mainly the refactoring of the software with a new, more flexible design and the addition of more choices for variograms and Kriging alternatives. Moreover, a component for the automatic leave-one-out error estimation was added. Therefore the required assessment was already performed and does not require to be done again. However, in "Design of the SIK package" section, besides the two modelling solution already presented in the previous manuscript, we added a further flow chart, which shows how to plug-in the SIK components to GEOframe-NewAGE.

(9) In some parts of the paper, it is specified that the data of 97 meteorological stations are used, but there are 93 stations in Appendix D. More than this, there are 81 stations in Figure 4; 57 stations in Figure 8. In short, an inconsistency in number of stations is in existence.

The total number of stations is 97. In Appendix, D 4 stations were missing for a mistake. All the stations are reported in figure 4 but due to visualization problems (the scale is too big), some of them overlap. The same visualization problems are present in figure 8, were some of the stations overlap.

(10) When examining Figure 8, I have doubts about a manipulation for high performance. Please enlighten me about this matter.

Regarding the manipulation for high performance, and also the doubts about the number of the stations, it is easy to check that nothing was made on the results, since all the simulations, inputs and outputs are available in the OSF project. We highly believe in the reproducibility of research and that's why we put the entire SIK project online. This helped also us to easily check and correct the mistake about the number of the stations.

(11) To get to the clarification of figures, I think Figure 3 and 5 must be made smaller, due to they are nonsensically very big.

We reduced the size of the two figures and we will further improve this aspect in the final version of the paper.

(12) In Figure 4, basin boundary and river network should be shown for more understanding of the study area. In the legend of Fig. 4, the ranges of elevation must be like "210.1-850, 850-1500, 1500-2100, 2100-2750, 2750-3388" with round numbers as an example.

The figure was changed accordingly to the suggestions.

(13) For Figure 5 and 9, please give an explanation of X and Y axis in the relevant places of the text.

The two explanations were added in the text.

(14) In the legend of Figure 5, please illustrate experimental semivariance as only black dot without line and the others as only line without dot.

The legend was correct.

(15) In Figure 6, size of dots can be smaller in terms of clearly view of each semivariogram model.

We tried to change the size of the dots but they overlaps anyway. Therefore, we decided to change the figure, showing only a comparison between the 4 types of Kriging, as suggested by the Reviewer #2.

(16) A scale bar (km) must be added on Figure 7 where the intervals of temperature must be as "-6-0, 0-5, 5-10, 10-15, 15-19" with round values.

The legend was correct and the scale bar added.

(17) Likewise, A scale bar and north arrow must be placed in Figure 8 which has two same legends of elevation, one of them must be removed.

The scale and the north were added.

(18) In comparison of spatial data, same color scale must be used for spatial layers. In this respect, Figure 7 (temperature scale) and 8 (elevation scale) are good examples. However, sizes of RMSE plots are not same in Figure 8. In despite of the authors allege the visualization reasons as excuse, I do not accept this situation. For a reasonable comparison, same size scale of RMSE must be used. I suggest the followed intervals for RMSE in both OK and DK output layers of June 2008. "0-1, 1-2, 2-3, 3-6, 6-9, >9 with max value". A conspicuous another issue about Figure 8 is that max RMSE value is 9 in spite of given as 11.95 in the text.

The figure was changed and also merged to figure 7, to shorten the Results section according to the Reviewer #2 suggestion.

(19) The measured rainfall is not shown as black dashed line in the legend of Figure 10.

Corrected

(20) In Figure 11, the color scale of precipitation is changed from white to dark blue. But the colors of precipitation classes are very close to each other, because of this situation the transition of precipitation surface is not distinguished exactly. To remove

this problem, selection of a more colorful scale will be suitable. For instance; red, yellow, sea green, light and dark blue, purple. It changes respectively from dry and wet. Two raster layers (DEM and precipitation), both of which contain white color, are overlapped using transparency feature of the layers in the same figure; but this is caused the confusion with regard to understanding of the figure. If the presentation of elevation layer is as contour lines, the confusion dies. The intervals of the precipitation and DEM must be respectively as "2.21-2.7, 2.7-3.2, 3.2, 3.7, 3.7-4.2, 4.2-4.7, 4.7-5.2, 5.2-5.7, 5.7-6.02" and "0-300, 300-2900, 2900-3388". Figure 11 also needs a scale bar.

The plot was changed accordingly to the reviewer suggestions.

(21) Together with NSE; R2, RMSE and PBIAS values of DK and similarly all the performance criteria of OK methods can be given in Figure 12.

According to the reviewer suggestions, we modified the plot and added also the R2, RMSE and PBIAS in the comparison between the two software.

(22) And all, the captions of the figures can be shortened as follows and the required explanations relating to the figures should be stated in the relevant places of the text. In all the figures, writings and titles of chart and axis must be had standard font size. Figure 1. Flow chart of modeling process. Figure 2. Flow chart of validation process. Figure 1. Flow chart of modeling and validation processes. (Figure 1 and 2 can be combined into one) Figure 4. Geo-location of study area and position of meteorological stations. Figure 5. Fitting of the experimental semivariogram for 15th June 2008 12:00 CET. Figure 6. Monthly variation of the NSE index over entire hourly temperature dataset. Figure 7. Maps and histograms of spatialized temperature for 15th February 2008 and 15th June 2008. Figure 8. Bubble plots of the RMSE obtained using OK and DK. Figure 9. Boxplots of the semivariograms of the precipitation event of 29th and 30th June 2008. Figure 10. Comparison between the four types of Kriging and the measured rainfall. Figure 11. Spatial interpolation of the precipitation applying OK and

the linear semivariogram model.

The captions were shortened according to reviewer suggestions

(23) A little more information such as slope, land cover, vegetation, hypsometric elevation and river discharge about the study area can be given. Basin characteristics are important in understanding and evaluation of hydro-meteorological modeling in particular type and distribution of precipitation for this study.

More information was added to the general description.

(24) Info of DEM is not presented in the paper. Please provide source of DEM.

The source of the DEM was added.

(24) Please give definitions, explanations and formulations of the performance criteria used (NSE, R2, RMSE, PBIAS) in the paper. For NSE, it is referred to Nash and Sutcliffe (1970).

- The definitions and references of the GOFs are now in a related Appendix.

(25) The headers of the tables can be changed as below because of they are also overlong. The statements about the tables can be presented in the relevant parts of the text not in the table header.

Changed accordingly

(26) It is emphasized that Bessel semivariogram model is the best in according to Table 1. How was this information reached? How was the integration of 4 criteria (NSE, R2, RMSE, PBIAS) achieved?

The integration of the 4 criteria was obtained using the R package "hydroGOF", which computes the GOFs between the measured and simulated values (in this case experimental semivariance and theoretical semivariance).

(26) Logarithmic model results also look pretty good, why was not the logarithmic model

in the best 5 of the authors? How were the best 5 determined? In my opinion, the logarithmic model is candidate for the best and Bessel, logarithmic, exponential, linear and spherical models are the best 5 semivariograms. Please clarify this matter.

This is true. The five best models are Bessel, logarithmic, exponential, linear and spherical. However, to be in line with the paper of Abera et al. (2017), and according to the suggestions of the Reviwer #2, we decided to keep only the results obtained using the Bessel model. In Abera et al. (2017), as well as in Haberlandt, 2007, it is stated that "the main difference is observed between the kriging methods, not between the semivariogram fitting within a single model". Therefore, we could have added also the results for the logarithmic model, but without obtaining valuable differences in the interpolation results. Nevertheless, the statement "All the models gave satisfactory results and, therefore, we chose to use the best 5: Bessel, Exponential, Gaussian, Linear and Spherical for the interpolation of the temperature dataset" was changed to " All the semivariogram models gave satisfactory results, with high values of NSE (0.72:0.92) and R2 (0.73:0.92), low values of RMSE (2.14 °C : 3.99 °C ) and PBIAS (-3.80% : -7.90%), confirming the accuracy of the calibration procedure. In order to assess the goodness of the of the interpolation, we performed the leave-one-out cross validation using the optimized hourly values of sill, nugget and range for the Bessel model, which is one of the best semivariograms according to the previous results."

(27) In Table 2, mean of performance criteria values of all the stations can be given for 4 types of Kriging.

According to the Authors, the Table is coherent with the text, since the results of the interpolation in the two station considered are compared.

(28) Table 1. Performance results of semivariogram models used Table 2. Results in terms of goodness of fit indices between the measured and interpolated rainfall values for two stations

Changed accordingly
(29) In the manuscript, it is stated that the overall performances of both SIK and gstat tools are good. I don't think gstat has good performance in according to Figure 12. Please research the success limits of NSE values in the literature. In this context, please rewrite the related parts and the last sentence of the abstract. In plain words, I did not expect it to be so different. SIK has almost overwhelming superiority. I have some suspicions in this regard. Please inform about the optimization of gstat, is it performed sensitively? How do you describe this failure and explain it?

We checked the results and we found an error in the comparison of the time series obtained with the two software (there was an inversion of the ID of two stations, which caused an offset of all the IDs). The plot was corrected and now the GOFs are really good for both software. The following part was added to the section: "Figure 12 shows the results of the temperature interpolation done with SIK and gstat, in terms of NSE, RMSE, PBIAS and R^2: the overall performances of both tools are very good. The NSE are always above the 0.65, while the RMSE are always lower then 2 °C. Figure 13 shows the results of the precipitation interpolation done with SIK and gstat, in terms of NSE, RMSE, PBIAS, R^2 and cumulative volume. Also in this case, both software are able to reproduce the rainfall event, well simulating the peaks. The results obtained for station ID 1152 are very good for both software, with a NSE >0.9. Both software show slightly worse results for station ID 2170, with lower values of NSE and R$^2$, higher RMSE, and an overestimation of the total rainfall (19 mm with $gstat$ and 20 mm with SIK, compared to the 16 mm recorded by the gauges)."

(30) In conclusion section, resolution of interpolation grid data (100 m) can be mentioned because of the resolution is an important issue for the hydrological model results.

Added

(31) The following abbreviations and symbols must be added to Appendix B: PSO: particle swarm optimization OK: ordinary kriging LOK: local ordinary kriging DK: de-

trended kriging LDK: local detrended kriging SIK: spatial interpolation kriging OMS3: object modeling system v.3 MS: modeling solutions DP: design patterns LOO: leave-one-out GIS: geographical information systems a.s.l. : above sea level CET: central European time DEM: digital elevation model DOI: digital object identifier $\Gamma$: matrix of the two point variograms $\Lambda$: N-uple of the unknown weights $\hat{\Sigma}$: N-uple containing the variograms among the ungauged site and the measured sites

Changed accordingly

(32) In most of the equations in Appendix C, multiplication is indicated by a dot. I think there is no need to use these dots.

All the dots were deleted.

(33) In Appendix D, sorting of the stations by elevation not station ID becomes more meaningful due to the strong relationship between the elevation and temperature or precipitation. Giving of mean and median elevation values of the all stations would be better.

All the info about the stations is going to be moved in a complementary material (according to the suggestions of the Reviewer #2).

(34) In appendix of code availability, the authors specify that the documentation of SIK components is available. Is a detailed user manual provided?

Yes. The link is present in the OSF project in the section "Documentation".

(35) The sentence of "Lastly, the authors declare that they have no conflict of interest." should be placed at the end of author contribution appendix.

The sentence was added.

Technical Corrections Corrections, typing errors

(36) In some parts of the paper, it is specified that the data of 97 meteorological stations
and 10 semivariograms are used, but there are 93 stations in Appendix D and 11 semivariogram models in Figure 5/Table 1.

The mistakes were corrected.

(37) P1 L7-P8 L14- P11 L8- P20 L18: "97" —> "93"

Changed accordingly

(38) P1 L5-P20 L20-P22 L2: "ten" —> "eleven"

Changed accordingly

(39) P10 L6-P10 L14-P11 in Fig. 5 caption-P11 L2-P12 in Table 1 header-P20 L2: "10" —> "11"

Changed accordingly

(40) P1 L3: "in a hydrological model" —> "to a hydrological model"

Changed accordingly

(41) P1 L8: "compared to the results" —> "compared to the results of temperature"

Changed accordingly

(42) P1 L11+: "These data, besides being important by themselves, are the natural input to distributed and semi-distributed hydrological models, where their quality and precision affect the accuracy of results." —> Please give a few references.

References were added accordingly.

(43) P1 L15+-P1 L19-P1 L22-P3 L5+-P5 L5-P7 L17: The references aren't chronologically presented, please chronologize.

Changed accordingly

(44) P1 L18: "anong" —> "among"

Changed accordingly

(45) P1 L20: "shown" —> "showed"

Changed accordingly

(46) P2 L25: "(Formetta et al., 2014).In order" —> "(Formetta et al., 2014). In order". No space after point.

Changed accordingly

(47) P3 L13-P3 L15-P3 L27: The ordering of in-equation parenthesis is not appropriate. The correct ordering is "{ [ ( ) ] }".

Changed accordingly

(48) P3 L9-P3 L22-P 3 L23: "N-ple" —> "N-uple" Changed accordingly

(49) P3 L26-P5 L21-P7 L11: "e.g." —> "e.g.," Changed accordingly

(50) P3 L27: ":=" —> "="

Changed accordingly

(51) P4 L2-P4 L3: "i.e." —> "i.e.,"

Changed accordingly

(52) P4 L19: "are the local" —> "are respectively the local"

Changed accordingly

(53) P4 L28: "interpolate a theoretical model for the semivariogram" —> "fit a theoretical model to the semivariogram"

Changed accordingly

(54) P4 L30: "produce maps" —> "produce continuous surface maps"

Changed accordingly

(55) P5 L7: "Efron and Efron" —> "Efron"

Changed accordingly

(56) P5 L10: "produced" —> "performed"

Changed accordingly

(57) P5 L16: "granularity whose rational" —> "detail whose logic"

Changed accordingly

(58) P5 L17: "that, for the reader convenience are presented in A" —> "presented in Appendix A for the reader convenience"

Changed accordingly

(59) P5 L19: "Object Modelling System v3 (OMS3)" —> "OMS3". Previously presented as open.

Changed accordingly

(60) P5 L20: "here" —> "are"

Changed accordingly

(61) P5 L22: "Modelling Solutions (MS)" —> "MS". Previously presented as open.

Changed accordingly

(62) P5 L22: "complicate" —> "complicated"

Changed accordingly

(63) P5 L25: "particular cares from the researcher who programs them" —> "any special effort of the computer programmer of them"

Changed accordingly

(64) P5 L29+: "in various components, and inserted in MS as alternatives, thus opening the way to compare, inside the same chain of tools" —> "various components inserted in MS as alternatives. Thus inside the same chain of tools, it enables to compare"

Changed accordingly

(65) P5 L31: "in Formetta et al. (2014) and in Bancheri (2017)" —> "Formetta et al. (2014) and Bancheri (2017)"

Changed accordingly

(66) P6 L12: "Particle swarm" —> "Particle swarm optimization (PSO) technique"

Changed accordingly

(67) P6 L16-P11 L7-P14 L2-P17 L2: "show" —> "shows"

Changed accordingly

(68) P7 L10: "addition(" —> " addition (". No space before parenthesis.

Changed accordingly

(69) P7 L11: "Design Patterns (DPs," —> "DP". Previously presented as open.

Changed accordingly

(70) P7 L13-P7 L14-P7 L15: "DPs" —> "DP"

Changed accordingly

(71) P8 L13-P20 L11: "modeling solutions" —> "MS". Previously presented as open.

Changed accordingly

(72) P9 in Fig. 4-P18 in Fig. 11: "Meteo_stations" —> "Meteorological stations" P9 in Fig. 4: "DEM" —> "DEM (m)" P18 in Fig. 11: "DEM" —> "Contour lines (m)". In Fig.

11, contour lines must be used instead of DEM.

Changed accordingly

(73) P9 L9: "out-layers" —> "outliers"

Changed accordingly

(74) P9 L10-P10 L3: "asses" —> "assess"

Changed accordingly

(75) P10 L1+: "nor too low or too high" —> "neither too low nor too high"

Changed accordingly

(76) P10 L10: "their performances" —> "performances"

Changed accordingly

(77) P11 L1: "the correlation coefficient" —> "the coefficient of determination"

Changed accordingly

(78) P11 L5: "4 typer of Kriging, OK, LOK, DK, LDK," —> "4 types of Kriging"

Changed accordingly

(79) P11 L9: It is specified that the two local cases were performed using the ten closest stations to the interpolation point. Why 10 stations? Has neighborhood analysis ever carried out?

The local cases have been performed using a fixed number of closest stations. In the case of temperature, this number was set equal to ten, since it was the right compromise between the mean distance among the stations and the number of recording stations for each time step.

(80) P11 L10: "meaning" —> "it means"

Changed accordingly

(81) P11 L12: "exists" —> "exist" Changed accordingly

(82) P12 in Table 1: Please sort the semivariograms by the performance.

We prefer to leave the alphabetical order of the semivariograms since, in this way, it is coherent with Figure 5 and Appendix E.

(83) P13 in Fig. 6: Please remove expression of "hourly temperature" from the chart titles as it is expressed in the caption.

Changed accordingly

(84) P14 L2: "7" —> "Figure 7"

Changed accordingly

(85) P14 L3: "during" —> "in"

Changed accordingly

(86) P15 L1: "alms" —> "m a.s.l."

Changed accordingly

(87) P15 L2: "1.83 oC" —> "1.83 oC in the DK method" Changed accordingly

(88) P15 in Fig. 8: Please remove expression of "monthly mean temperature" from the map titles.

Changed accordingly

(89) The altitude legend scale has not min and max values, please add these.

Changed accordingly

(90) P16 in Fig. 9 chart title: "30 h" —> "11 h"

Changed accordingly

(91) P16 L2: "953 m" —> "943 m". According to Appendix D.

Changed accordingly

(92) P16 L6: "local ordinary kriging" —> "LOK". Previously presented as open.

Changed accordingly

(93) P16 L7: "ID 2170" —> "ID 1270" Changed accordingly

(94) P16 L6-P16 L8: It is stated that the LOK computed using the 5 closer stations. Is it determined with manual calibration of the neighborhood? If search of neighbor numbers was performed, it must be specified in the text. If not, explain that how it was determined. In addition to, the number of closest neighbor stations is 10 for temperature and 5 for rainfall. Please interpret this.

In the case of precipitation, the number of closest stations was set equal to five, given the prevalent convective nature of summer precipitation phenomena and the lower number of active gauge stations for each time step. A description of the analysis of the data, the treatment of the outliers, the choice of the number of the bins and of the closest stations is reported in the section Setup.

(95) P17 L1: "DEM (100 m resolution)" —> "DEM"

Changed accordingly

(96) P17 L3: "[mm/h]" —> "(mm/h)"

Changed accordingly

(97) P19 L4: "design pattern" —> "DS". Previously presented as open.

Changed accordingly

(98) P19 L4: "figure 3" —> "Figure 3"

Changed accordingly

(99) P19 L7: "opertion" —> "operation"

Changed accordingly

(100) P19 L10: "offers" —> "offer"

Changed accordingly

(101) P19 L11+: "the same number of bins and the same cutoff distance" —> "the same number of bins and cutoff distance"

Changed accordingly

(102) P19 in Fig. 12 legend: "SI component" —> "SIK component"

Changed accordingly

(103) P19 in Fig. 12 caption: "package" —> "packages"

Changed accordingly

(104) P19 in Fig. 12: NSE value 1.0 is not found in the figure.

Changed accordingly

(105) P20 L13: "design patterns" —> "DS". Previously presented as open.

Changed accordingly

(106) P20 L18+: "Both the interpolation of the temperature and the interpolation of the rainfall" —> "Both the interpolation of the temperature and the rainfall"

Changed accordingly

(107) P20 L22: "detrended Kriging" —> "DK". Previously presented as open.

Changed accordingly

(108) P20 L22: "Single event rainfall, on the contrary do not" —> "On the contrary, single event rainfall did not"

Changed accordingly

(109) P20 L26: "GPL v3" —> "GPL v3 (www.gnu.org/licenses/gpl-3.0.en.html)"

Changed accordingly

(110) P20 L27: "a a-personal" —> "a-personal"

Changed accordingly (111) P20 L28: "thus"

Changed accordingly

(112) P21 L1: "instead, for instance, of Mercurial," —> "instead of for instance Mercurial"

Changed accordingly

(113) P21 L1: "found at" —> "found at followed link"

Changed accordingly

(114) P21 L5: "includes" —> "include"

Changed accordingly

(115) P21 L7: "Maven and Gradle, in particular" —> "in particular Maven and Gradle"

Changed accordingly

(116) P21 L7+: "Our favor, among the possibilities, was for" —> "Our favor among the possibilities was"

Changed accordingly

(117) P21 L12: "instead than another All of them" —> "than another. All of them"

Changed accordingly

(118) P21 L13: "programmers' community, but are" —> "community of programmers, but they are"

Changed accordingly

(119) P21 L13+: "scientists making increasingly difficult for them maintain their own code" —> "scientists who are increasingly struggling against difficulties to maintain their own code"

Changed accordingly

(120) P21 L18: "and run all the software" —> "and all the software"

Changed accordingly

(121) P21 L20: "GPL v3 license (www.gnu.org/licenses/gpl-3.0.en.html)" —> "GPL v3 license"

Changed accordingly

(122) P21 L22: "alternatives are, among others," —> "alternatives among others are"

Changed accordingly

(123) P21 L25: "our" —> "ours"

Changed accordingly

(124) P22 in Appendix B: "Percent bias" —> "percent bias"

Changed accordingly

(125) P22 in Appendix B: "Coefficient of determination" —> "coefficient of determination"

Changed accordingly

(126) P22 in Appendix B: "Root mean square error" —> "root mean square error"

Changed accordingly

(127) P25 in Appendix D: The elevation of station 90145 is 3399 m as given in Appendix D and P15 L1. However the max elevation value of study area is 3388 m in Fig. 4. Here, there is a mistake (3399>3388), please correct it.

Changed accordingly

(128) P29 L5: "Fisheries research" —> "Fisheries Research"

Changed accordingly

(129) P29 L25-P29 L37-P30 L10: "Oxford university press" —> "Oxford University Press"

Changed accordingly

(130) P29 L29: "Efron, B. and Efron, B."

Changed accordingly

(131) P29 L36: "Springer Publishing" must be added.

Changed accordingly

(132) P30 L2: "Journal of hydrology" —> "Journal of Hydrology"

Changed accordingly

(133) P30 L7: "Journal of applied meteorology" —> "Journal of Applied Meteorology"

Changed accordingly

(134) P30 L8: "International journal of geographical information systems" —> "International Journal of Geographical Information Systems" Changed accordingly

(135) P30 L21: "Biotechnol. Agron. Soc. Environ., vol. 17(2), 392-406" must be added.

Changed accordingly

(136) P30 L27: "Mathematical geology" —> "Mathematical Geology"

Changed accordingly

(137) P30 L31: "Publ. Climatol., 46" must be added.

Changed accordingly

(138) P30 L35: "JAWRA Journal of the American Water Resources Association, 21(3), 365–380" must be added.

Changed accordingly

(139) P30 L36: "Monthly weather review" —> "Monthly Weather Review"

Changed accordingly

(140) The references in P30 L22 and P30 L24 must be replaced due to the chronological order.

The order depends on the second author name.

(141) P31 L3: "Agricultural systems" —> "Agricultural Systems"

Changed accordingly

(142) P31 L6: "Data Acquisition And Processing, Analysis, Forecasting And Other Applications, (WMO No.168)" must be added.

Changed accordingly

Recommendations

The validation of SIK component can be performed with remote sensed data (radar and satellite products) besides of ground based data (earth observation stations).

The climatic and environmental variables can be analyzed at different time scales as

daily, monthly, seasonal and annual.

The interpolation performance can be evaluated in different areas in point of multi-site approach.

Co-Kriging using secondary data as elevation, vegetation and temperature can be embodied into the SIK.

Kling-Gupta Efficiency (KGE) criteria whose advantages over NSE are stated in Gupta et al., 2009, can be added as performance benchmark.

Interpolated temperature and rainfall data can be inputted into a hydrological model. And accuracy and reliability of interpolation can be tested in terms of performance of hydrological model.

In assessment of interpolation performance, jack knife and k-fold cross validation can be utilized for independent data set of validation procedure.

Fit of normal distribution of residuals can be tested due to Kriging modeling is a stochastic process.

Uncertainty analysis can be carried out. Comparison based on pixels of raster layer can be implemented.

Comparison in terms of lowland-highland, underestimate-overestimate, density of stations and representation of low-high rainfall.

All the previous recommendations are great ideas for future versions of the software and we put it in the wish list of improvements to do, in the Github repository of the component. Various case studies using remote sensed data have been already performed using the GEOframe-NewAGE infrastructure and SIK performances will be going to appear also in future papers. The software has been also intensively used for educational purposes with students of the Hydrology class in Trento, a sort of a crash test, which was successfully accomplished. The integration of Co-Kriging, given the software de-
sign of the SIK package, is easy and straightforward, and the interested researcher could find the "How to do" in our repository in Github. However, these are not efforts we can afford at present without delaying too much the final review of this paper, whose scope has, as well as all the papers, a limited scope.

References Adhikary, S. K., Muttil, N. and Yilmaz, A. G., 2016. Genetic programming-based ordinary kriging for spatial interpolation of rainfall. J. Hydrol. Eng., 21 (2), -1–1. Bostan, P. A., Heuvelink, G. B. M. and Akyurek, S. Z., 2012. Comparison of regression and kriging techniques for mapping the average annual precipitation of Turkey. Int. J. of Applied Earth Observation and Geoinformation, 19, 115-126. Di Piazza, A., Lo Conti, F., Noto, L. V., Viola, F. and La Loggia, G., 2011. Comparative analysis of different techniques for spatial interpolation of rainfall data to create a serially complete monthly time series of precipitation for Sicily, Italy. Int. J. of Applied Earth Observation and Geoinformation, 13, 396-408. Gupta, H. V., Kling, H., Yilmaz, K. K. and Martinez, G. F., 2009. Decomposition of the mean squared error and NSE performance criteria: Implications for improving hydrological modelling. Journal of Hydrology, 377, 80-91. Jin Q., Zhang J., Shi M. and Huang J., 2016, Estimating Loess Plateau average annual precipitation with multiple lineer regression kriging and geographically weighted regression kriging. Water, 8, 266, 1-20. Krige, D. G., 1951. A statistical approach to some mine valuations and allied problems at the Witwatersrand. MSc Thesis of the University of Witwatersrand. Nash, J. E., and Sutcliffe, J. V., 1970. River flow forecasting through conceptual models. Part I-A discussion of principles. Journal of Hydrology, 10(3), 282-290. Saghafian, B. and Bondarabadi, S. R., 2008. Validity of regional rainfall spatial distribution methods in mountainous areas. J. of Hydrol. Eng., 13 (7), 531-540. Wang, S., Huang, G. H., Lin, Q. G., Li, Z., Zhang, H. and Fan, Y. R., 2014. Comparison of interpolation methods for estimating spatial distribution of precipitation in Ontario, Canada. Int. J. Climatol., 34: 3745-3751.

―――――――――――――――――――――――

---

## Author Comment (AC2) · 19 Apr 2018

R1: The technical paper presents a package for the interpolation of climatological variables using kriging techniques. The structure of the package is described, and the results of the application to two datasets (a rainfall and a temperature one) are presented. The authors also provide a comparison with the R package gstat. The technical paper aims at describing a package that could be of interest for the geoscientific community due to the fact that it is specifically designed for the interpolation of climatological variables and is suitable to be easily merged into hydrological models. The manuscript then would fit the purposes of the journal, if focused on the description of this package. Despite this, the aim of the paper is not clear. On the one hand, reading the abstract, the expected focus is the tool itself: its capability and the novelty it introduces in comparison with the available software. On the other hand, going through the paper one can realize that just a limited part of the work is focused on the package, and the core of it is focused on the analysis of the results obtained with the different tested kriging methods. Considering that a wide geostatistical literature focused on the assessment of the quality of the interpolation of climatological variables already exists (e.g., Hartkamp et al., 1999, Moral F. J., 2010, etc.), and that the analysis of the performance of the package are limited to sect.6, I think that the authors should adjust the focus of the whole manuscript (or at least clarify their aims) before it can be accepted for publication.

A1: The authors thank the reviewer for her/his observations, which helped us improve our manuscript. This response does not yet produce a new manuscript, which will be submitted after the closure of the review phase, upon request of the Editor. We introduce here, however, the main adjustments that we will make in the final version of the revised paper based upon the reviewer's suggestions. According to the suggested reviews, we will try to focus more on the package and on its core and shorten the analysis of the results. According to what is stated in the abstract, we will try to respect the purpose of the study, which are: (1) to present a geostatistical software easy to use and easy to plug-in to a hydrological model, (2) to show a practical example of an accurately designed software in the perspective of reproducible research, (3) to show the goodness of the software applications, in order to have a reliable alternative to other traditionally used tools. Moreover, we will try to make a better comparison between SIK and other available software, especially with R gstat.

R2: A series of major comments follows, all related to this topic, and some minors ones are added, that I think the authors should address before the manuscript is ready for publication. Major comments Most of the manuscript is dedicated to the kriging methodology. The use of kriging methodologies is quite established in the hydrological literature and many studies on the performance of the different kriging methods can be found. The core part of the paper should be the description of the tool and the

assessment of its potentialities.

A2: According to the reviewer suggestion, we decided to move all the equations related to the Kriging theory to the Appendix A and to leave in the text only the most important references. In this way, the core of the paper is going to be the section of the design of the package and the assessment of its potentialities.

R: In the detail:

1. My opinion is that the authors dedicate too much space at the description of the theoretical basis of the kriging (Sect. 2) while some references and a general outline would be enough.

A1. As stated in the previous answer, we moved all the equations of section 2 to a related appendix, leaving only few references in the main text.

2. The analysis of the results of the case study turns out to be the core of the paper (Sect. 5 is the longest one with 6 out of 12 figures dedicated). Section 5 should be revised, shortened, and the number of related figures should be reduced. The interest of the case study is in checking the functionality of the package rather than to carry out an analysis of the performances of the various interpolators

A2. According to the reviewer suggestions, we decided to modify the figures, in order to present only three figures for each case (temperature and rainfall): the fitting of the experimental semivariogram, the leave-one-out results using a single semivariogram model and the raster interpolation, using a single semivariogram model, overlapped to the bubble plot of the errors. For example, Figure 6 now shows only the comparison of the interpolations between the 4 types of Kriging, using the Bessel semivariogram model.

3. While the structure of the tool is clearly explained in sect. 3 the analysis of the performance, compared to other software, is not sufficient. As the authors themselves state, it is an important point "to justify the introduction of an alternative software".

The authors just refer to general "way longer" (L.5 P.19) computational time of gstat compared to SIK, and a comparison of the NSE for a single model (figure 12). This does not appears enough for a scientific paper: the authors should provide numbers and figures to support and highlight performances of their package in comparison with other software. Their opinion of "greater ease" (L.8 P.19) is not enough, considering that gstat turns out to be more versatile under some points of view (e.g., it considers both omnidirectional and directional variograms) and that there are some other kriging packages offering a user-friendly and easy-to-manage interface (e.g., the v.kriging command in GRASS-Gis).

A3. We agree that previous analysis could appear not too much accurate for a scientific paper. However, a quantitative comparison of the two tools could be unfair (for gstat, at the end). SIK and gstat have different objectives indeed. The software gstat is suitable for use in simple tasks with short scripts, because of its C computational core, embedde in R which is flexible and easy to use. It is, instead, hard to plug-in to a hydrological model, which involves temporal and spatial steps (because R coded for-loops are really slow, being an interpreted language). Besides implementing modelling solutions (MS) which mix for instance Java and R is possible (for instance our new version of the OMS console can do it - at scripting level) but it cannot be considered a clean solution for building an operational tool (R binding into OMS was mostly intended to be used in the analysis of results, more that in computing them). SIK, instead is suitable to fit into large and complicate modeling solution because it is fully OMS-compliant and because of the JVM performance optimizations that comes in when the same algorithm is executed several times (like the kriging interpolation for a year of hourly time steps). However, we changed the text adding more information about differences and similarities between the two softwares. Moreover, we modified the plot related to the temperature comparison (there was also mistake) and we added a new plot on rainfall interpolation, with further information about the comparison. The following part was added to the text of the new manuscript: "A comparison between SIK and the R package $gstat$ was made in order to highlight their differences and

similarities, and to justify the deployment of an alternative software. Benchmarks or quantitative performance comparisons would not have been useful or completely truthful since the "velocity" of computation (a classic quantitative comparison) depends on too many factors, some of which are described below. Therefore, we performed a qualitative comparison between the two softwares accounting for design, the implemented features, and the accuracy of the results. In our opinion, the two tools that we analyzed have different purposes. This can be seen just by looking at the features of the relative programming languages. The gstat software is developed in C with a part of the code in R language. It must be executed using the various R environments. SIK is developed in Java (7) as a group of OMS components and it can be executed form within the OMS console, as a stand-alone Java programs, or embedded in other codes in languages that support Java bindings. Java is slower than 3rd generation languages such as C. However, in the course of Java development several optimizations, such as "just-in-time compilation" and "adaptive optimization", have been introduced to improve the performance of its Java Virtual Machine (JVM). These techniques identify recurrently executed algorithms, so called "hot spots", and dynamically recompile them at run-time. Eventually, the hot spots gain valuable computational speed. C is one of the fastest compiled languages. But only the computational core of gsta$ is coded in C; the management of temporal steps, such as "for-loops", and data structures must be scripted in R. Undoubtedly, R is a very powerful programming language, mainly because of its flat learning curve, and easy syntax and semantics, but it is fully interpreted, which makes it very slow. As a result, the comparison of the speed of computation for a single temporal Kriging interpolation is unfair against Java, since the JVM cannot exploit its optimization tools for a single computation. On the other hand, the comparison of the speed of computation for a year of hourly Kringing interpolations is biased against R, because temporal steps affect most of the computational time. In terms of functionality, gstat computes both omnidirectional and directional semivariongrams, while SIK does not implement directional semivariograms yet (although we have included this feature on the software wish list). Furthermore, gstat provides four more

theoretical semivariogram models with respect to SIK, these are: Matern, Matern with Stein's parameterizations, Wave, and Legendre. Adding the desired theoretical model to any SIK-TV component would be easy and straightforward, thanks to the DP implemented, as shown in Figure 2, but they are not available at present. Regarding the estimates that the two packages offer, these are usually different. Comparisons were made with both the temperature and rainfall datasets used in section Setup. Semivariograms were computed using the same number of bins and cutoff distance. Figure 12 shows the results of the temperature interpolations done with SIK and gstat, in terms of NSE, RMSE, PBIAS and Rˆ2: the overall performances of both tools are very good. The NSE values are always above the 0.65, while the RMSE are always lower then 2 °C. Figure 13 shows the results of the precipitation interpolation done with SIK and gstat, in terms of NSE, RMSE, PBIAS, Rˆ2 and cumulative volumes. Also in this case, both softwares are able to reproduce the rainfall event well, simulating the peaks. The results obtained for station ID 1152 are very good for both softwares, with a NSE >0.9. Both softwares show slightly worse results for station ID 2170, with lower values of NSE and R$ˆ2$, higher RMSE, and an overestimation of the total rainfall (19 mm with gstat and 20 mm with SIK, compared to the 16 mm recorded by the gauges). In conclusion, gstat is a powerful, flexible tool to get fast results with fast scripting in answer to single, specific questions (with some implementation efforts user-side); SIK is a tool that is ready to be integrated into broader MS, specifically because of its OMS-compliant design. The interpolations of both temperature and rainfall confirm the quality and accuracy of the predictions obtained using the SIK package, demonstrating that it is a good competitor of R gstat."

R: Minor comments 4. Abstract: the abstract should be reviewed according to the modification of the structure of the paper. At the moment it does not describe the focus of the work.

A4. We revised the structure of the paper, trying to follow the three purposes stated in the abstract, according to the reviewer suggestions.

5. L.9, P.2 – It must be pointed out that GRASS-GIS is also available as an open-source GIS offering a user-friendly (v.kriging) function for performing kriging

A5.    In the Introduction chapter, according also to the reviews of the Referee #1, we added some missing software that was worth mentioning, among which GRASS-GIS. Now a better comparison of SIK with the existing open-source software is present.    The following part was added to the main text: "Several geostatistical tools are made available to the scientific community.    Among them, PyKrige (https://github.com/bsmurphy/PyKrige), SAGA GIS kriging (www.saga-gis.org), GRASS (grass.osgeo.org), Surfpack (https://dakota.sandia.gov/content/surfpack), R gstat (www.cran.r-project.org) and the High Performance Geostatistics Library HPGL (https://www.github.com/hpgl). However, only few of them could be considered as alternatives of SIK, i.e. the ones that are open source, comprehensively documented and actively developed: – Dakota (Surfpack): C++ software with flexible interface that provides optimization algorithms, uncertainty quantification, parameter estimation, and sensitivity analysis for supporting computational models and simulators (Adams et al., 2009); – PyKrige: python package that allows for both 2D and 3D ordinary and universal Kriging computation with flexible design for custom variogram implementation (Murphy, 2014); – gstat: R package (computational core coded in C) that supports block kriging, simple, ordinary and universal (co)kriging and many other features (Pebesma, 2004), (Gräler et al., 2016). It is historically the leading software in this field. While GIS-based tools, such as QGIS and GRASS (v.kriging) Krigings, are easily included into scripts leveraging GIS capabilities, they are not easily included into complicated MS. As well as being open-source, SIK is the only Java-based and component-based software of those mentioned above. Moreover, it implements a quick way to plug-in to hydrological models and automatic calibration algorithms. We decided to compare the performances of SIK and R gstat, since the latter is one of the most widely used tools in the scientific community."

6. Variables should be named always in italic (e.g., "B" L.22, P.3), and measurement

units not in italic (e.g., km2 L.1, P.9)

A6. Changed accordingly

7. Figure 1 and 2 can be merged together as they differ only for the last block. 8.

A7. Figure 1 was changed accordingly.

8. APPENDICES: I think the authors make a misuse of the appendices. Appendix A could be merged in the main part of the paper as deal with an important phase of the development of the package and Appendix D is almost useless for the aims of the paper (the table should be provided as supplementary material). Reducing the number of the symbols also Appendix B would be useless, and the meaning of the symbol could be inserted along the main text.

A8. All the Appendices were revised according to the reviewer's suggestions. In particular: - Appendix A now introduces the Kriging equations; - Appendix B contains the list of Acronyms; - Appendix C contains the description of the GOFs; - Appendix D contains the list of semivariograms models implemented in SIK.

9. Writing style and use of English could be improved: the manuscript would benefit from extensive English editing by a native speaker. The text should also be double-checked for typos and other minor errors (a non-exhaustive list follows): o L.18, P.1. "anong" -> "among" o L. 8, P.2.

A9. The English is going to be improved in the revisions and also the spelling errors.

10. There is a useless open bracket o L. 4, P. 4.

A10. Changed accordingly

11. "Appendix A" -> "Appendix C" o L. 17, P. 5

A11. Changed accordingly

12. I don't think "contemporary" is a proper word in this case o L. 10, P. 8

A12. Changed accordingly

13. "classes" -> "class" o L. 14, P. 8

A13. Changed accordingly

14. " . . . in D" -> "..in Appendix D" o L. 7, P. 10

A14. Changed accordingly

15. "where" -> "were" 10.

A15. Changed accordingly

16. Reference style is not uniform, sometimes brackets include only the year (e.g., L.1, P.2) and sometimes name and year (e.g., L.25, P.2).

A16. We will try to check all the reference styles.

17. Please adopt the journal style.

A17. We will adopt the journal style.

References Hartkamp, A. D., De Beurs, K., Stein, A., & White, J. W. (1999). Interpolation techniques for climate variables.

Moral, F. J. (2010). Comparison of different geostatistical approaches to map climate variables: application to precipitation. International Journal of Climatology, 30(4), 620-631

---

## Referee Report (RR1)

Reviewer Report

**"The design, deployment and testing of Kriging models in GEOframe"**
by Marialaura Bancheri et al.

The manuscript has significantly improved with respect to the first version. Now its aims are clearer and the overall structure more consistent. I still would like to see more comparison with the available alternatives, but I understand there are technical problems in pursuing these kind of comparison.

Despite the improvements, I think minor revision are still required before publication on any international journal.

A couple of technical aspects need first to be clarified:

1. L1-3P14: How do you choose the "best variogram model"? Is it an automatic on manual procedure? Based on what? The authors remarks in the conclusions that "the tests also show how it is possible to choose between 11 variograms…" but the choosing rationale is not explained across the manuscript.
2. L21P19: "Regarding the estimates… these are usually different". I can not understand how it is possible. If the setup of the methodology is the same, the variogram bins and distances the same and, obviously, the kriging equation the same, how can the two packages provide different results?

In addition, despite the authors' revision, my greatest concerns are still related to style and use of English language. I am not English mother tongue either, but I strongly suggest the authors to undergo a global language and style revision with the help of an expert (mother tongue, if possible) as wording and language structures not commonly used in English are still present, especially in some sections of the manuscript: they severely degrade the readability of the paper. The consistence with the journal style has to be checked too.

In the following, please find some examples of the language, structure and style problems. It is just a non-exhaustive list:

3. L8P1: "For both variables" → "For both the variables"
4. L22P1: "…by several authors, (Tabios and Salas, 1985; Jarvis and Stuart, 2001), among others" → I don't think this is consistent with the citation stile you adopted. As (Tabios and Salas, 1985; Jarvis and Stuart, 2001) are part of the sentence they should be cited, e.g., like the authors at L23P1 "…by several authors, among others Tabios and Salas (1985) and Jarvis and Stuart (2001), who…" or you can rephrase: "…by several authors ( e.g., Tabios and Salas, 1985; Jarvis and Stuart, 2001), who…"
5. L15P2: "make" → "makes"
6. LL24-26P2: "In this work…error of estimates" → The sentence is not well structured, please rephrase it (consider using a bulleted list)
7. L14P3: "First some preliminary…" → "First, some preliminary…"
8. L20P3: "…measured data, (…)." → "…measured data (….).". Please, verify all across the manuscript mis-uses of commas in the correspondence of brackets.
9. L6P4: "…too strict, (…)." → "…too strict (…)."
10. L19P4: "…of errors, (…)." → "…of errors (….)."
11. LL26-27P4: "Further details… Appendix A" → Already mentioned at the beginning of the section, consider removing.
12. L28P4: "Design of the SIK package, its deployment and use cases" → The title sounds really bad in English, consider alternatives, e.g., "Design, deployment and use cases of the SIK package" or "The SIK package: design, deployment and use cases"
13. L29P4: "…and on the use cases…" → "…and of the use cases…"

14. L30P4: "…four levels of detail, the logic of which is explained below" → It is not clear what lever of detail you are referring to and I can't find where you explain their logic. Do you refer to the 4 OMS3 components? I think the can not be defined "levels of details" as they are just the necessary steps for the kriging algorithm to work. Please, try to adopt a more readable structure, where the 4 levels are clearly listed and explained. Otherwise, please remove the sentence.

15. FIGURE 1 → Why don't you call the OMS components with the name you assign them in the manuscript instead of the titles? It would help readability (Krigings → SIK-K, etc.).

16. L6P6: "…the four OMS components…" → What about the "Particle Swarm"? Isn't it an OMS component? In the case the components are five.

17. L12P6: "…the use of DP" → Please explain the acronyms the first time you use them. Here and across the manuscript.

18. L15P6: "…makes the code easier to "read" and maintain." → "…makes the code easier to be read and maintained."

19. L18P6: "…of good practice can be found in the scientific literature, (…)." → "…of good practice can be found (…)." (it is clear from the previous lines you refer to scientific literature).

20. L1P9: "Here we delineate the practices implemented in building the SIK package that make it a Reproducible Research System…" → "Here, we delineate the practices implemented in building the SIK package for making it a Reproducible Research System…"

21. FIGURE 3 – caption: "…as used in (Bancheri, 2017)" → "…as used in Bancheri (2017)" (see previous comments)

22. LL3-8P9: "Although the initial code (let's call it v 0.1) was already available from a control version system under GPL v3 license (www.gnu.org/licenses/gpl-3.0.en.html), the repository was owned by the original Author. An non-personal repository was judged to be better suited to host a collaborative work. Therefore, for SIK and its companion tools the collective GEOframe organization repository was created under Github (www.github.com), using Git (www.git-scm.com), and can be found at the following link (www.github.com/ geoframecomponents)." → Please, correct the typos: "Despite the initial code, that will be referred to as v0.1, was already available from a control version system under GPL v3 license (www.gnu.org/licenses/gpl-3.0.en.html), the repository was owned by the original author. A non-personal repository was judged to be better suited for hosting a collaborative work. Therefore, for SIK and its companion tools the collective GEOframe organization repository was created under Github (www.github.com), using Git (www.git-scm.com). It can be found at the following link: www.github.com/geoframecomponents)."

23. LL1-27P10: all this part of the section requires a deep revision as there are significant phrasing and structural errors that make it hardly readable, I just report some examples:
    o L1: "Moreover, code v0.1" → Remove "moreover", the sentence is not related to the previous
    o L1: "contemporary" → Here "modern" is better.
    o L2: "various libraries that concur" → "various concurring libraries"
    o L5: "Maven and Grandle can…" → "Both Maven and Grandle can…"
    o L6: "…depending on use of the….with respect to the…" → "…thanks to the use of the…compared to the…."
    o L7: "…has the practical effect of abstracting…" → "…allows at abstracting…"
    o L9: "…support Gradle and Maven (and Ant)…" → "…support both Gradle, Maven and Ant…"
    o L10: "….but they are hardly used at all by scientists…" → "…but rarely by scientists…"
    o L11: "…researchers can…" → "…researchers could…"
    o L13-14: "while certainly not necessary to doing good science" → Please remove the part, as it just complicates the structure of the sentence without adding anything (or, at least, correct "to doing" with "for doing").
    o LL17-22: "Travis CI, (https://travis-ci.org), is a good choice for a continuous integration service, which uses GitHub as a web-based git repository hosting service. Continuous integration, (Meyer, 2014), is the practice of merging all developer working copies to a

shared mainline several times a day. Unit Tests, (Beck, 2003), are built with the code and run each time the merging is done. The continuous integration service automatically builds the executable codes, checks if the tests are performed correctly and returns a positive answer if all is done properly. Eventually, major code commits are tagged with release numbers, under the GPL v3 license." → The structure is confused: first you have to explain that continuous integration is, then you can suggest a good choice. Moreover, it is not clear if you effectively used Travis CI. Please, clarify (and, again, be careful to the use of commas): "Continuous integration (Meyer, 2014), is the practice of merging all developer working copies to a shared mainline several times a day. Unit Tests (Beck, 2003) are built with the code and run each time the merging is done. The continuous integration service automatically builds the executable codes, checks if the tests are performed correctly and returns a positive answer if all is done properly. Eventually, major code commits are tagged with release numbers, under the GPL v3 license. For this purposes, we chose/not choose to use Travis CI (https://travis-ci.org), which uses GitHub as a web-based git repository hosting service, and is a good choice for a continuous integration service."

- LL26-27P10: "This could prove important…in a paper where relevant results were obtained and, perhaps, it could… within research groups, at least for ours". → "This could be important… in a paper and, perhaps, it could… within research groups." (Try to keep the sentence simple: all the paper should report significant results, and it's clear that it is your opinion).

24. L17P11: "…any outliers from the dataset" → "…any outliers.".
25. L8P12: "…all 11 theoretical models" → "…all the 11 theoretical models."
26. L21P12: "15th June 2008" → "15 June 2008", all across the manuscript. Journal guidelines suggest to use the format: dd month yyyy. The "th" is not required.
27. "…the semivariance [$h$] and the distance in meters respectively" → "…the semivariance [h] and the distance in [m] respectively"
28. L5P13: "…with high values…" → "…with large values…"
29. L6P13 "3.99°C" → "3.99 °C"
30. L4P14: "…shows the results between the measured and interpolated values using the four…" → "…shows the results for the four…"
31. L12P14: "…model" → "…models." ; "15th February" → "15 February"
32. L14-15P14: "…the biggest error…the biggest error…" → "…the largest error…the largest error…"
33. L7P15: [$h$] → [h]
34. FIGURE 9 – caption: "29th" "30th" → "29" "30" (also on L7P17)
35. L2P18: "…biggest.." → "..largest…"
36. LL4-6P18+LL1-3P19: "A comparison between SIK and the R package gstat was made in order to highlight their differences and similarities, and to justify the deployment of 5 an alternative software. Benchmarks or quantitative performance comparisons would not have been useful or completely truthful since the "velocity" of computation (a classic quantitative comparison) depends on too many. factors, some of which are described below. Therefore, we performed a qualitative comparison between the two softwares accounting for design, the implemented features, and the accuracy of the results. In our opinion,…" → I suggest to change the order of the sentence: "A comparison between SIK and the R package gstat was made in order to highlight their differences and similarities, and to justify the deployment of 5 an alternative software. We performed a qualitative comparison between the two softwares accounting for design, the implemented features, and the accuracy of the results. Benchmarks or quantitative performance comparisons would not have been useful or completely truthful since the "velocity" of computation (a classic quantitative comparison) depends on too many factors, some of which are described below. Moreover, in our opinion, …."
37. L5P19: "…it can be executed form within…" → "…it can be executed within…"
38. LL16-18P19 (and across the manuscript): please, adopt a consistent style for "*gstat*".
39. L19P19: "…to SIK, these are: Matern,…" → "…to SIK: Matern,…"
40. L21P19: "not available at present" → "not available yet"

41. L3P21: I think "To test the performance…variables" (LL3-6P22) should follow here (after "4 types of Kriging interpolations") as it it a part of the "summary" of the paper to be inserted befor the general comments on the results.
42. L12P21: I think it is true "only under certain conditions" and need to be clarified.
43. L7-9P22: I would remove the sentence "As expected,…. Elevation", as it is an aspect out of the aims of the paper, it is an "expected" result and does not fit the conclusion section.
44. L1P25: "Indexes of goodness of fit" → "Goodness of fit indices"
45. L4P26: "..the eleven theoretical…" → "…the 11 theoretical…"

---

## Author Response (AR2)

Answer to the Reviewer Report
**"The design, deployment and testing of Kriging models in GEOframe"** by Marialaura
Bancheri et al.

The manuscript has significantly improved with respect to the first version. Now its aims are
clearer and the overall structure more consistent. I still would like to see more comparison
with the available alternatives, but I understand there are technical problems in pursuing these
kind of comparison.
Despite the improvements, I think minor revision are still required before publication on any
international journal.

*The Authors thank the reviewer for his recognitions and for the reviews.*

A couple of technical aspects need first to be clarified:
Q1. L1-3P14: How do you choose the "best variogram model"? Is it an automatic on manual
procedure? Based on what? The authors remarks in the conclusions that "the tests also show
how it is possible to choose between 11 variograms…" but the choosing rationale is not
explained across the manuscript.

A1. *The author ranked the semivariogram models according to the GOF shown in Table 1.
The procedure was manual and performed using the R package hydroGOF. This is specified
in L1-7P13.*

Q2. L21P19: "Regarding the estimates… these are usually different". I can not understand
how it is possible. If the setup of the methodology is the same, the variogram bins and
distances the same and, obviously, the kriging equation the same, how can the two packages
provide different results?

A2. *The differences in the performances are due to the  solver of the linear system of the
weights of the Kriging component. Moreover, these differences are larger in the month were
more no-values are present in the time series of the interpolated variables. The treatment of
the no-values is going to be improved in the next versions of the SIK package.*

Q: In addition, despite the authors' revision, my greatest concerns are still related to style and
use of English language. I am not English mother tongue either, but I strongly suggest the
authors to undergo a global language and style revision with the help of an expert (mother
tongue, if possible) as wording and language structures not commonly used in English are
still present, especially in some sections of the manuscript: they severely degrade the
readability of the paper. The consistence with the journal style has to be checked too.

A: *The paper was already revised by an English mother tongue, before the submission of the
reviews. However, the authors will go through  a further check, as  asked by the reviewer.*

In the following, please find some examples of the language, structure and style problems. It is just a non-exhaustive list:

3. L8P1: "For both variables" → "For both the variables"

*Changed accordingly.*

4. L22P1: "…by several authors, (Tabios and Salas, 1985; Jarvis and Stuart, 2001), among others" → I don't think this is consistent with the citation stile you adopted. As (Tabios and Salas, 1985; Jarvis and Stuart, 2001) are part of the sentence they should be cited, e.g., like the authors at L23P1 "…by several authors, among others Tabios and Salas (1985) and Jarvis and Stuart (2001), who…" or you can rephrase: "…by several authors ( e.g., Tabios and Salas, 1985; Jarvis and Stuart, 2001), who…"

*The sentence was changed in: "Their performances have been assessed by several authors, among others  Tabios and Salas (1985) and Jarvis and Stuart (2001), who..."*

5. L15P2: "make" → "makes"

*Changed accordingly.*

6. LL24-26P2: "In this work…error of estimates" → The sentence is not well structured, please rephrase it (consider using a bulleted list)

*A list of the component was introduced. The sentence was changed in:*
*"In this work, the SIK package is presented as four components:*
   *1. the first is used for the production of the experimental semivariograms;*
   *2. the second is used for the production of the theoretical semivariograms;*
   *3. the third is used for the Kriging interpolation;*
   *4. and the last is used for an automatic and easy jackknife resampling to asses the error of estimates. "*

7. L14P3: "First some preliminary…" → "First, some preliminary…"

*Changed accordingly.*

8. L20P3: "…measured data, (…)." → "…measured data (….)."". Please, verify all across the manuscript mis-uses of commas in the correspondence of brackets.

*Changed accordingly.*

9. L6P4: "…too strict, (…)." → "…too strict (…)."

*Changed accordingly.*

10. L19P4: "…of errors, (…)." → "…of errors (….)."

*Changed accordingly.*

11. LL26-27P4: "Further details… Appendix A" → Already mentioned at the beginning of the section, consider removing.

*The sentence was removed.*

12. L28P4: "Design of the SIK package, its deployment and use cases" → The title sounds really bad in English, consider alternatives, e.g., "Design, deployment and use cases of the SIK package" or "The SIK package: design, deployment and use cases"

*Changed in  "Design, deployment and use cases of the SIK package".*

13. L29P4: "…and on the use cases…" → "…and of the use cases…"

*Changed accordingly.*

14. L30P4: "…four levels of detail, the logic of which is explained below" → It is not clear what lever of detail you are referring to and I can't find where you explain their logic. Do you refer to the 4 OMS3 components? I think the can not be defined "levels of details" as they are just the necessary steps for the kriging algorithm to work. Please, try to adopt a more readable structure, where the 4 levels are clearly listed and explained. Otherwise, please remove the sentence.

*The sentence was changed in : "On the basis of the analysis of the mathematical problems and of the use cases delineated in the previous section, the design of the software was organized in four OMS3 components, the logic of which is explained below."*

15. FIGURE 1 → Why don't you call the OMS components with the name you assign them in the manuscript instead of the titles? It would help readability (Krigings → SIK-K, etc.).

*Figures 1, 3 and 4 were changed and now the components are called with the same names assigned in the manuscript.*

16. L6P6: "…the four OMS components…" → What about the "Particle Swarm"? Isn't it an OMS component? In the case the components are five.

*The components of the SIK package are actually four. The particle Swarm is a calibrator integrated in the core of OMS3.*

17. L12P6: "…the use of DP" → Please explain the acronyms the first time you use them. Here and across the manuscript.

*The first time the author introduced the acronyms DP is before in L30P2, and there DP is defined. Besides, it also appears in the list of acronyms in appendix B..*

18. L15P6: "…makes the code easier to "read" and maintain." → "…makes the code easier to be read and maintained."

*Changed accordingly.*

19. L18P6: "…of good practice can be found in the scientific literature, (…)." → "…of good practice can be found (…)." (it is clear from the previous lines you refer to scientific literature).

*Changed accordingly.*

20. L1P9: "Here we delineate the practices implemented in building the SIK package that make it a Reproducible Research System…" → "Here, we delineate the practices implemented in building the SIK package for making it a Reproducible Research System…"

*Changed accordingly.*

21. FIGURE 3 – caption: "…as used in (Bancheri, 2017)" → "…as used in Bancheri (2017)" (see previous comments)

*Changed accordingly.*

22. LL3-8P9: "Although the initial code (let's call it v 0.1) was already available from a control version system under GPL v3 license (www.gnu.org/licenses/gpl-3.0.en.html), the repository was owned by the original Author. An non-personal repository was judged to be better suited to host a collaborative work. Therefore, for SIK and its companion tools the collective GEOframe organization repository was created under Github (www.github.com), using Git (www.git-scm.com), and can be found at the following link (www.github.com/ geoframecomponents)." → Please, correct the typos: "Despite the initial code, that will be referred to as v0.1, was already available from a control version system under GPL v3 license

(www.gnu.org/licenses/gpl-3.0.en.html), the repository was owned by the original author. A non-personal repository was judged to be better suited for hosting a collaborative work. Therefore, for SIK and its companion tools the collective GEOframe organization repository was created under Github (www.github.com), using Git (www.git-scm.com). It can be found at the following link: www.github.com/geoframecomponents).”

*The typos were corrected and the sentence was changed according to the reviewer suggestions.*

23. LL1-27P10: all this part of the section requires a deep revision as there are significant phrasing and structural errors that make it hardly readable, I just report some examples:

o L1: “Moreover, code v0.1” → Remove “moreover”, the sentence is not related to the previous
o L1: “contemporary” → Here “modern” is better.
o L2: “various libraries that concur” → “various concurring libraries”
o L5: “Maven and Grandle can…” → “Both Maven and Grandle can…”
o L6: “…depending on use of the….with respect to the…” → “…thanks to the use of the…compared to the….”
o L7: “…has the practical effect of abstracting…” → “…allows at abstracting…”
o L9: “…support Gradle and Maven (and Ant)…” → “…support both Gradle, Maven and Ant…”
o L10: “….but they are hardly used at all by scientists…” → “…but rarely by scientists…”
o L11: “…researchers can…” → “…researchers could…”
o L13-14: “while certainly not necessary to doing good science” → Please remove the part, as it just complicates the structure of the sentence without adding anything (or, at least, correct “to doing” with “for doing”).
o LL17-22: “Travis CI, (https://travis-ci.org), is a good choice for a continuous integration service, which uses GitHub as a web-based git repository hosting service. Continuous integration, (Meyer, 2014), is the practice of merging all developer working copies to a shared mainline several times a day. Unit Tests, (Beck, 2003), are built with the code and run each time the merging is done. The continuous integration service automatically builds the executable codes, checks if the tests are performed correctly and returns a positive answer if all is done properly. Eventually, major code commits are tagged with release numbers, under the GPL v3 license.” → The structure is confused: first you have to explain that continuous integration is, then you can suggest a good choice. Moreover, it is not clear if you effectively used Travis CI. Please, clarify (and, again, be careful to the use of commas): “Continuous integration (Meyer, 2014), is the practice of merging all developer working copies to a shared mainline several times a day. Unit Tests (Beck, 2003) are built with the code and run each time the merging is done. The continuous integration service automatically builds the executable codes, checks if the tests are performed correctly and returns a positive answer if all is done properly. Eventually, major code commits are tagged with release numbers, under the GPL v3 license. For this purposes, we chose/not choose to use Travis CI (https://travisci.org), which uses GitHub as a web-based git repository hosting service, and is a good choice for a continuous integration service."
o LL26-27P10: "This could prove important…in a paper where relevant results were obtained and, perhaps, it could… within research groups, at least for ours". → "This could be important… in a paper and, perhaps, it could… within research groups." (Try to keep the sentence simple: all the paper should report significant results, and it's clear that it is your opinion).

*All the typos and the sentences were corrected in the manuscript according to the reviewer suggestions.*

24. L17P11: "…any outliers from the dataset" → "…any outliers.".

*Changed accordingly.*

25. L8P12: "…all 11 theoretical models" → "…all the 11 theoretical models."

*Changed accordingly.*

26. L21P12: "15th June 2008" → "15 June 2008", all across the manuscript. Journal guidelines suggest to use the format: dd month yyyy. The "th" is not required.

*Changed accordingly.*

27. "…the semivariance [$h$] and the distance in meters respectively" → "…the semivariance [h] and the distance in [m] respectively"

*Changed accordingly.*

28. L5P13: "…with high values…" → "…with large values…"

*Changed accordingly.*

29. L6P13 "3.99°C" → "3.99 °C"

*Changed accordingly.*

30. L4P14: "…shows the results between the measured and interpolated values using the four…" → "…shows the results for the four…"

*Changed accordingly.*

31. L12P14: "…model" → "…models." ; "15th February" → "15 February"

*Changed accordingly. Model refers to the Bessel semivariogram.*

32. L14-15P14: "…the biggest error…the biggest error…" → "…the largest error…the largest error…"

*Changed accordingly.*

33. L7P15: [*h*] → [h]

*Changed accordingly.*

34. FIGURE 9 – caption: "29th" "30th" → "29" "30" (also on L7P17)

*Changed accordingly.*

35. L2P18: "…biggest.." → "..largest…"

*Changed accordingly.*

36. LL4-6P18+LL1-3P19: "A comparison between SIK and the R package gstat was made in order to highlight their differences and similarities, and to justify the deployment of 5 an alternative software. Benchmarks or quantitative performance comparisons would not have been useful or completely truthful since the "velocity" of computation (a classic quantitative comparison) depends on too many. factors, some of which are described below. Therefore, we performed a qualitative comparison between the two softwares accounting for design, the implemented features, and the accuracy of the results. In our opinion,…" → I suggest to change the order of the sentence: "A comparison between SIK and the R package gstat was made in order to highlight their differences and similarities, and to justify the deployment of an alternative software. We performed a qualitative comparison between the two softwares accounting for design, the implemented features, and the accuracy of the results. Benchmarks or quantitative performance comparisons would not have been useful or completely truthful since the "velocity" of computation (a classic quantitative comparison) depends on too many factors, some of which are described below. Moreover, in our opinion, …."

*The Authors agree with the reviewer and the order of the sentences was changed according to the reviews.*

37. L5P19: "…it can be executed from within…" → "…it can be executed within…"

*Changed accordingly.*

38. LL16-18P19 (and across the manuscript): please, adopt a consistent style for "*gstat*".

*All the "gstat" were corrected in the text.*

39. L19P19: "…to SIK, these are: Matern,…" → "…to SIK: Matern,…"

*Changed accordingly.*

40. L21P19: "not available at present" → "not available yet"

*Changed accordingly.*

41. L3P21: I think "To test the performance…variables" (LL3-6P22) should follow here (after "4 types of Kriging interpolations") as it it a part of the "summary" of the paper to be inserted before the general comments on the results.

*The Authors agree with the reviewer and the order of the sentences was changed according to the reviews.*

42. L12P21: I think it is true "only under certain conditions" and need to be clarified.

*It was added in the sentence.*

43. L7-9P22: I would remove the sentence "As expected,…. Elevation", as it is an aspect out of the aims of the paper, it is an "expected" result and does not fit the conclusion section.

*The sentence was removed.*

44. L1P25: "Indexes of goodness of fit" → "Goodness of fit indices"

*Changed accordingly.*

45. L4P26: "..the eleven theoretical…" → "…the 11 theoretical…"

*Changed accordingly.*